# Supraglacial lake bathymetry automatically derived from ICESat-2 constraining lake depth estimates from multi-source satellite imagery

Rajashree Tri Datta[1,2,3] and Bert Wouters [4,5]

[1] Earth Systems Science Interdisciplinary Center, University of Maryland, College Park, MD
[2] NASA Goddard Space Flight Center, Greenbelt,MD
[3] Department of Atmospheric and Oceanic Sciences, University of Colorado Boulder, Boulder, CO, USA
[4] Department of Physics, Institute for Marine and Atmospheric Research, Utrecht University, Utrecht, NL
[5] Faculty of Civil Engineering and Geosciences, Delft University of Technology, Delft, NL

*Correspondence to*: Rajashree Tri Datta (Tri.Datta@gmail.com)

**Abstract.** We introduce an algorithm (*Watta*), which automatically calculates supraglacial lake bathymetry and potential ice layers along tracks of the ICESat-2 laser altimeter. *Watta* uses photon heights estimated by the ICESat-2 ATL03 product and extracts supraglacial lake surface, bottom, depth corrected for refraction and (sub)surface ice cover in addition to producing surface heights at the native resolution of the ATL03 photon cloud. These measurements are used to constrain empirical estimates of lake depth from satellite imagery, which were thus far dependent on sparse sets of in-situ measurements for calibration. Imagery sources include Landsat 8 OLI, Sentinel-2 and high-resolution Planet Labs PlanetScope and SkySat data, used here for the first time to calculate supraglacial lake depths. The *Watta* algorithm was developed and tested using a set of 46 lakes near Sermeq Kujalleq (Jakobshavn) glacier in Western Greenland, and we use multiple imagery sources (available for 45 of these lakes) to assess the use of the red vs green band to extrapolate depths along a profile to full lake volumes. We use *Watta*-derived estimates in conjunction with high-resolution imagery from both satellite-based sources (tasked over the season) and nearly-simultaneous Operation IceBridge CAMBOT (Continuous Airborne Mapping By Optical Translator) imagery (on a single airborne flight) for a focused study of the drainage of a single lake over the 2019 melt season. Our results suggest that the use of multiple imagery sources (both publicly-available and commercial), in combination with altimetry-based depths, can move towards capturing the evolution of supraglacial hydrology at improved spatial and temporal scales.

## 1 Introduction

Ice loss from Greenland and Antarctica is the greatest current contributor to rising sea levels, and paleodata and modelling efforts indicate that enhanced mass loss of these ice sheets may become irreversible if certain major tipping points are passed (IPCC, 2019). Recent observations have shown that ice
loss is accelerating faster than projected (Slater et al., 2018), with a sixfold increase since the 1970s/1980s. In Antarctica, this was  largely driven by increased ocean melting of outlet glaciers

(Rignot et al., 2019), while on the Greenland Ice Sheet mass loss is further promoted by increased surface melt and runoff (Mouginot et al., 2019).

Owing to the non-linear relationship between increasing summer air temperatures and surface melt (Trusel et al., 2018), meltwater production has increased rapidly on the Greenland Ice Sheet (van
den Broeke et al., 2016). In the summer of 2019, advection of warm, wet mid-latitude air led to a summer mass loss unprecedented in the past 50 years, with widespread surface melt occurring up to the highest regions of the ice sheet (Tedesco and Fettweis, 2020; Sasgen et al., 2020). Concurrent with the increase in melt extent and duration, supraglacial lakes - which form when meltwater runoff collects in local topographic lows - are now a common feature on large parts of the ice sheets and have become
more extensive and have advanced inland toward higher elevations in the past decades (Gledhill and Williamson, 2018; Leeson, 2015; Howat et al., 2013)

These meltwater lakes and streams are a key component of the hydrological system of both ice sheets. Hydrological systems over Antarctic ice shelves have been identified as a major factor for potential ice shelf collapse (Bell et al., 2018). While mass loss in Antarctica over the next 100 years is
generally thought to be dominated by the basal melt under ice shelves (Schlegel et al., 2018), emerging research has focused on the potential importance of surface hydrology over Antarctica (Arthur et al., 2020). Supraglacial lakes have been observed around the margin of the Antarctic Ice Sheet up to high elevations (Stokes et al,, 2019) and are likely to become more prevalent on firn-depleted ice shelves in future warming scenarios, which could potentially trigger their collapse and consequently lead to
accelerated sea level rise (Lai et al., 2020). Meltwater pathways can include surface flow into lakes and then streams, leading, in Greenland, to direct loss to the bed from lake drainage or the sudden termination of a stream into a moulin, or near-surface flow where ice slabs can limit vertical motion (MacFerrin et al., 2019). The complex links between supraglacial hydrological systems and englacial or subglacial pathways can potentially be deduced by capitalizing on increasingly higher-resolution
imagery and classification techniques of supraglacial feature types (Yang et al., 2017). Past remote-sensing work has derived lake volumes from high-resolution (~1 m) Worldview imagery using a physical optical depth approach as well as an empirical method using in-situ estimates (Moussavi et al., 2016; Pope et al., 2016). Recent work has developed an automated algorithm applying the physically-based method to Landsat 8 and Sentinel 2 to track specific hydrological features and quantify the
seasonal evolution of surface hydrology (Dell et al., 2020). The physical method assumes that wind-driven surface waves are minimal, that the slope of lake bottoms are gradual and that lake-bottom albedos are homogoneous (Sneed and Hamilton, 2011). The empirically-based method was first applied by Box and Ski (2007) using MODIS imagery and advanced by Legleiter et al. (2014), which used high-resolution WorldView 2 imagery. Both the physically-based and empirically-based methods are
limited to supraglacial lakes which contain minimal particulate matter (Arthur et al., 2020), and by the depth of the lake, assuming that the reflection depletion in imagery is limited at great depths, implying a physical limit to the ability to calculate depth (Box and Ski, 2007). Pope et al. (2016) estimates that the greatest depth that could be calculated by these methods was 5 m. Additional work has applied a similar physically-based approach using Sentinel-2 from Copernicus (Williamson et al., 2018), Landsat 7 and 8
(Banwell et al., 2014), Aster imagery (Sneed and Hamilton, 2007) and a combination of Landsat and Sentinel-2 imagery (Moussavi et al., 2020). Although Sentinel-2 provides relatively high resolution (10

m) imagery with substantial coverage at a 4-day to weekly interval, usable imagery is often limited by cloud-cover, and the resolution of small streams and ice cover is imperfect. Commercial satellite imagery, which is poised to expand substantially in the future, can help fill the gap in coverage of small-scale melt and melt-induced features at a higher spatial (<3 m) and temporal (multiple daily passes) resolution, complementing estimates resolved from Sentinel-2.

The ICESat-2 laser altimeter data, available since 2018, has now introduced the potential to replace the *in situ* measurements used in empirical (supraglacial lake depth) bathymetric methods with satellite laser bathymetric depths at a high vertical resolution, consequently extracting lake volumes from imagery (Parrish et al., 2019; Albright and Craig, 2020; Thomas et al., 2020). Here, we present a new algorithm, titled "*Watta*", using the ICESat-2 laser altimeter to derive properties of supraglacial lakes. *Watta* was first presented in Fricker et al. (2020), demonstrating both the potential for ICESat-2-based bathymetry estimates and the greater accuracy of empirically-based lake depths in comparison to physically-based estimates; the latter tended to underestimate lake depth by over 2 m. However, we note that physically-based methods have the advantage of being dependent on imagery alone. In addition to bathymetry derived from the difference between the air-water and water-ice interface, this algorithm assigns a probability for surface type characteristics to photon returns along-track. These types include lakes, refrozen lakes, lakes with ice layers on top as well as under the surface. *Watta* also returns surface heights at the native resolution of the ATL03 photon cloud (0.7 m), allowing the algorithm to capture small-scale changes in surface relief when multiple passes are differenced. Additionally, we exploit a range of imagery data to validate the surface types and to derive spectrally-driven depth estimates calibrated to ICESat-2-based depths, thereby providing an estimate for meltwater volume over the full image. We compare empirically-based volume estimates derived from a single ICESat-2 based depth estimate but from multiple imagery sources with different spatial resolutions (and without an atmospheric correction) to better understand the importance of spatial resolution and radiometric calibration on the relative accuracy of depth volume estimates.

The method is tested and refined using representative sections along the flowline of Sermeq Kujalleq (Jakobshavn Isbræ), one of the fastest-moving glaciers in Greenland, as well as the slower-flowing Sarqardliup Sermia. The repeat-tasking of Planet SkySat imagery was designed to coincide with ICESat-2 tracks (Fig. 1), capturing lake depths at various stages of lake development during the unusually intense melt season of summer of 2019 (Tedesco and Fettweis, 2020). One of the major motivations for this tasking effort was its coincidence with several NASA Operation IceBridge (OIB) flights at the beginning and end of the summer. Data from multiple instruments aboard OIB could potentially provide additional insight in future work, and within this study, we use OIB CAMBOT (Continuous Airborne Mapping by Optical Translator) imagery as a part of a focused multi-instrument study of the evolution of a supraglacial lake. The availability of simultaneous laser altimetry and high-resolution imagery over the season provided a rich test dataset with which to extract altimetry-based estimates of supraglacial lakes at various points in the season. Here, we present initial results exploiting this dataset as well as introducing the *Watta* ICESat-2 surface feature detection algorithm.

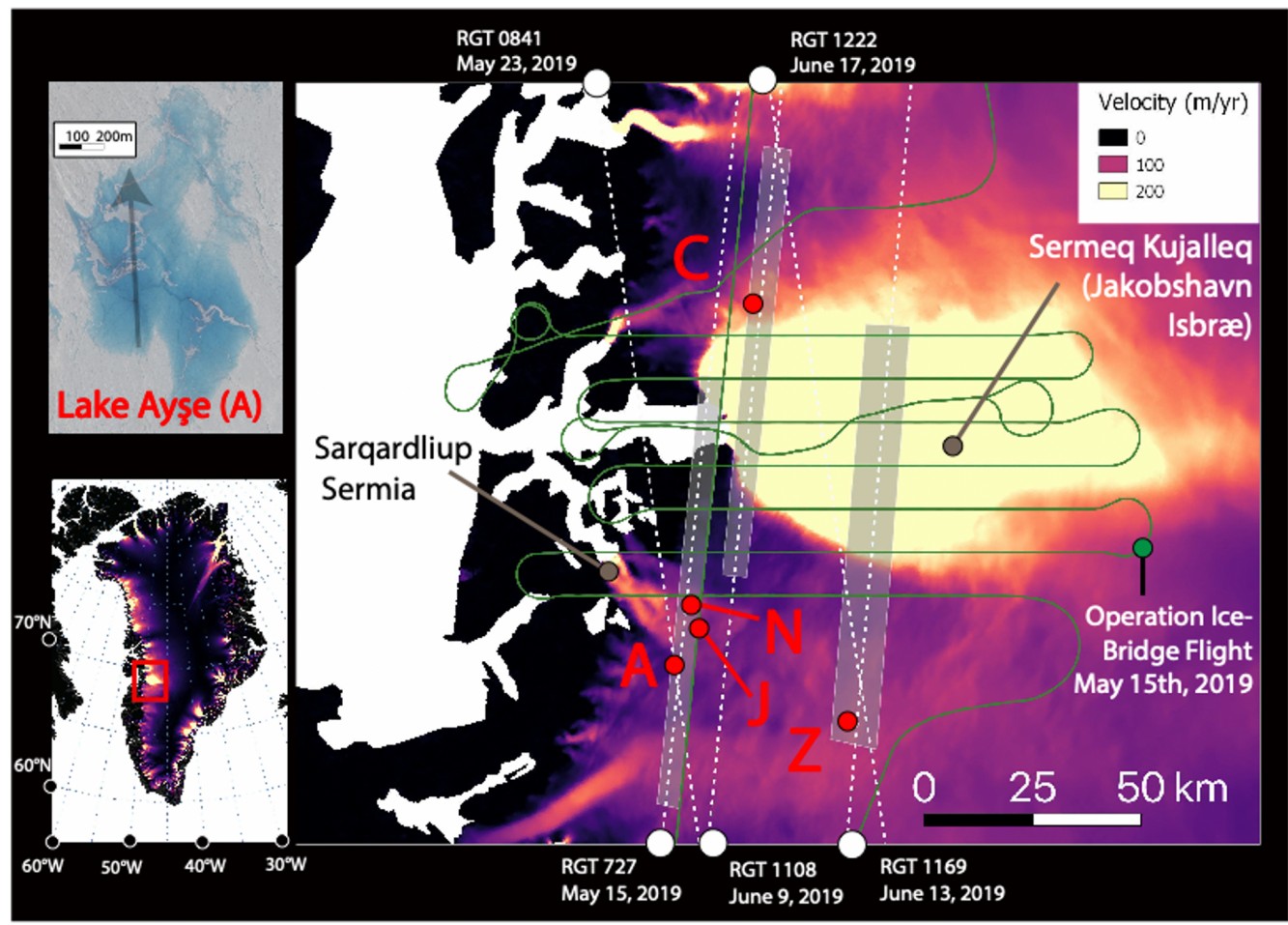

**Figure 1: Study region over Sermeq Kujalleq and Sarqardliup Sermia. Top Left: Lake Ayşe on May 23rd using Planet SkySat visual imagery, Bottom left, study region over Western Greenland. Right: Main region with repeat-tasking locations for Planet SkySat shown in grey boxes over annual velocity estimates from MEaSUREs (NSIDC). Center tracks ICESat-2 reference ground tracks shown in white. Operation IceBridge flight on May 15th, 2019 shown in green. Five lakes indicated in red discussed throughout text include C: Lake Cecily, A: Lake Ayşe, J: Lake Julian, N: Lake Niels, Z: Lake Zadie**

## 2 Data Sources

### 2.1 Satellite Altimetry

Our *Watta* method relies on individual photon heights as measured by ICESat-2's (Ice, Cloud, and Land Elevation Satellite) ATLAS (Advanced Topographic Laser Altimeter System) instrument, distributed in the ICESat-2 ATL03 product, L2A, Global Geolocated Photon Data (Neumann et al., 2019). The polar orbiting ICESat-2 satellite was launched in September, 2018 to continue the mission begun by ICESat

(2003-2009) and bridged with the airborne Operation IceBridge mission, namely to provide ice sheet mass balance estimates at an unprecedented level of accuracy. ATLAS is a photon-counting 532 nm laser altimeter aboard ICESat-2 split into 6 beams which are divided into 3 pairs (separated by 3.3 km), where beams within each pair are separated by 90 m. Within a single track, the beam pair is designated by a number, i.e. "3" in "gt3r". Each beam pair consists of a strong and weak beam, with the strong
beam returning 0.6-3.9 signal photons per laser pulse vs 0.6-1.0 signal photons per laser pulse for the weak beam (Neumann et al., 2019). The beam is designated with "r" or "l", depending on the orientation of the satellite, as in "r" in "gt3r".

While the strong beam produces a stronger signal, we have developed the *Watta* algorithm to work effectively with both strong and weak beams. ATL03 produces a photon cloud where each photon
is geolocated to within a 6.5 m accuracy and a footprint size of ~11m (Magruder et al., 2021) with an associated height as well as a confidence level [high, medium or low], and is produced at an along-track horizontal resolution of 0.7 m. While the ATL06 product (Smith et al., 2019) provides highly-accurate surface height estimates at a coarser resolution, the higher spatial resolution of the ATL03 product can be used to deduce fine-scale surface characteristics, as with the *Watta* algorithm. Over water bodies,
ICESat-2 can produce returns both over the surface over the lake as well as the bottom of the lake (Fair et al., 2020; Fricker et al., 2020; Parrish et al., 2019); these dual returns are used by *Watta* to extract supraglacial lake depths, as well as lake surface characteristics.

**2.2 High-resolution imagery near Sermeq Kujalleq**

For imagery sources, in addition to freely-available Landsat OLI (30 m) and Sentinel-2 (10 m) imagery, we incorporate very high resolution imagery from Planet Labs, including Dove-R (3 m) and SkySat (~1 m). The latter is used to validate surface types, while all imagery sources are used to derive spectrally-driven depth estimates calibrated to ICESat-2-based depths. Additionally, the high spatial resolution of SkySat imagery allows for the identification of small-scale features on the surface and bottom of
supraglacial lakes, which we use to interpret the temporal evolution of lake characteristics in a number of case studies. SkySat imagery did not include an atmospheric correction, and we therefore used TOA (Top of the Atmosphere) reflectance values from Landsat, Sentinel-2 and SkySat imagery to calculate supraglacial lake depth for the sake of consistency. PlanetScope Dove-R data provided surface reflectance values only and is known to have issues with radiometry (Saunier et al., 2020). However,
because the method used here derives lake depth values empirically (rather than physically), this work presents the opportunity to develop accurate depth estimates using high-resolution data where calibration is imperfect, but where the data availability is high. This is particularly true for data from the PlanetScope constellation, which are frequently captured multiple times within a single day. Relative spectral response curves for the bands used in this study are red, blue and green and NIR as shown in
Fig. S1b. Finally, all imagery was coregistered with ICESat-2 using the GIMP-2 digital elevation model (DEM), which has a vertical accuracy (as compared to ICESat), within ±1 m over most ice surfaces and ±30 m over areas with high relief (Howat et al., 2014) and is used for the geolocation of Landsat imagery (detailed in Section 3.2).

As a part of this project, SkySat imagery was tasked for repeat cycles of ~4 days over the 2019
Greenland melt season in selected locations, producing usable imagery at varying intervals based on
cloud cover. Each of the 3 areas of interest presented here were approximately 600 km$^2$. Repeat imagery
was specifically chosen to cover flowlines of fast-flowing glaciers, including Sermeq Kujalleq, as in
this study (Fig. 1). In addition, repeat tracks were designed to coincide with both (a) overpasses of the
recently-launched NASA ICESat-2 laser altimeter and (b) several flights of the airborne NASA
Operation IceBridge (OIB) mission in the beginning and end of the season. Here, we present the first
work exploiting this stacked dataset for method development, restricted to available satellite
imagery/altimetry. We note that for Lake Julian, discussed in section 5, OIB conducted a flight on
2019/5/15, thus capturing observations from multiple instruments onboard OIB, including CAMBOT
imagery and the Airborne Topographic Mapper (ATM). While ATM-based lake depth estimates could
potentially be compared to the lake depths calculated from the near-simultaneous ICESat-2 overpass,
this is outside the scope of this study. We discuss Lake Julian in detail in order to facilitate potential
future research at this site.

      The final set of lakes used for the development of the *Watta* method included 50 lakes captured
by ICESat-2 (46 over Sermeq Kujalleq and Sarqardliup Sermia, and 4 additional lakes in the southwest,
not shown here), 14 of which coincided with very high-resolution imagery (SkySat) within a 3-day
window. The date/times/imagery IDs/ICESat-2 details for all data sources are presented in
Supplemental Table 1.

### 3 Methods

We derive supraglacial lake volume from a given imagery source in four steps. We first calculate lake
depths along an ICESat-2 beam using the *Watta* algorithm applied to the ICESat-2 ATL03 photon
cloud. Secondly, we coregister *Watta*- based surface and lake bottom heights with the imagery source
(itself co-registered to a common Landsat base) and delineate lake boundaries in the process (methods
detailed in Section 3.2). Finally, we develop an empirical relationship between ICESat-2 based depths
and coincident imagery which can be applied to calculate lake depths over the full image. The empirical
relationship is based on the exponential decay of reflectance at water depth, as detailed by Box and Ski
(2007). In the original work, *in situ* depth estimates (D) and reflectance values from imagery (R) were
used to estimate the α-coefficients in eq. 1, which were then applied to calculate water depth over the
full-scale of imagery where lakes are delineated:
$$D = \alpha_0 / (R + \alpha_1) + \alpha_2 \quad \text{(eq. 1)}$$
However, such *in situ* estimates are scarce in space and time, and here we exploit the direct depths from
the *Watta* algorithm to derive time, location and sensor specific estimates of the α-coefficients.

### 3.1 Watta

*Watta* is an algorithm which takes ICESat-2 ATL03 photon data as input and automatically detects
supraglacial surface features with an associated likelihood. In its current state, the algorithm detects
lakes and their associated surface, lake bottom and corrected depth estimate as well as subsurface ice

when present (Fig. 2). We also exploit the algorithm for the detection of frozen streams in this study. The codebase for *Watta* is divided into a module which calculates surface and bottom returns ("Surface Detection") at the native 0.7 m resolution of ICESat-2, and a second "Interpretive" module that resolves the calculated bottom/surface to specific supraglacial features, in this case lakes. The Surface Detection
module determines, for a collection of photons surrounding any individual photon (75 collected before and 75 collected after; selected in step a), heights with the three strongest peak probabilities within in a kernel density (step b). This provides estimates for (1) a height for the surface or top of a lake or refrozen pond (2) a lake bottom and (3) a third height value, which can potentially be subsurface ice (Fig. 2 step a). We note that photons are selected without regard to ATL03 confidence level. Although
the bin width (and therefore vertical resolution) used to calculate heights is 0.1 m, we perform a second kernel density estimate calculation using a 0.3 m bin width to confirm robustness of the initial bottom estimate, i.e. that a coarser calculation produces a bottom height near that of the finer-resolution (0.1 m bin width) calculation. In post-processing step c, outliers are identified in comparison to surface/bottom heights within a larger horizontal window, whereby the number of standard deviations used to detect an
outlier and the number of photons used to calculate a mean (window) increase over several steps. Where outliers are found, the kernel density estimate (steps a,b) are recalculated with a larger number of photons (in multiples of 75) to account for any erroneous calculations generated by insufficient photon density. Where values continue to be outliers, they are removed from the estimates to be interpolated instead. The final output of the Surface Detection module include a calculated surface and potential lake
bottom return at the native resolution of the ATL03 photon cloud.

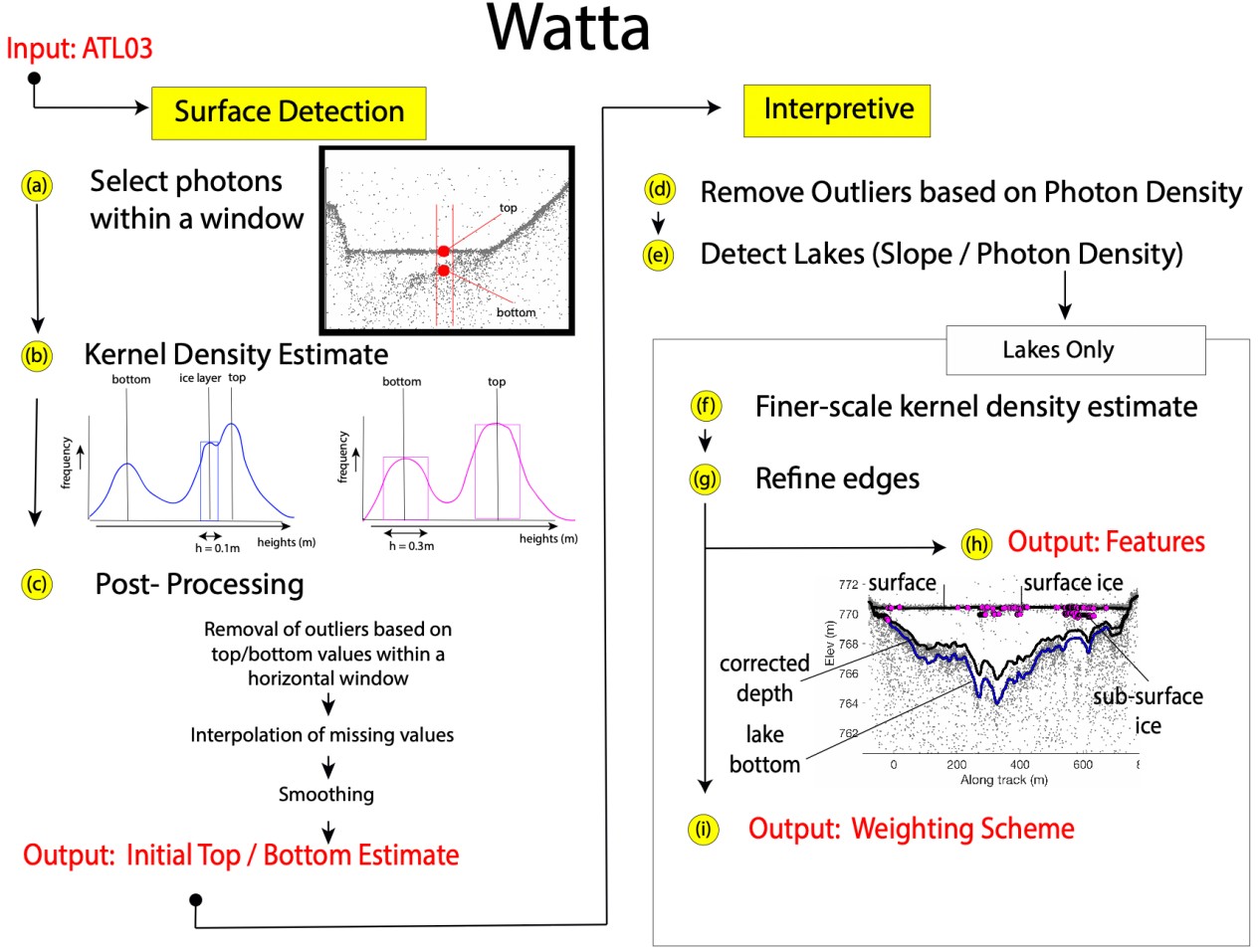

**Figure 2. Diagram of the *Watta* methods described in main text.**

The Interpretive module uses output from the Surface Detection module to automatically determine locations of surface features (e.g. lakes or frozen streams) as well as characteristics of a lake, e.g. the presence of refrozen ice at the surface. First, remaining outliers are removed by calculating a local background and surface photon density for each ATL03 photon, as determined from a 5000 photon-count window surrounding the estimate location (step d). We then remove those estimated heights for

top (surface) and bottom (potential lake bottom) where the photon density more closely resembles the background photon density than a surface density estimate. In step e, we detect breaks in the slope of the top (surface) to divide the satellite pass into segments which are potential lakes. For example, a semiparabolic depression in topography, with a high absolute value of the slope, is broken by a lake surface, where the slope approaches zero (e.g. as for the lake shown in Fig. 2 step a). For each of these

potential lakes, we then perform several steps to both refine lake surface/bottom and to assign the lake a

class based on its properties. To produce a lake bottom value with greater accuracy, we first perform a recalculation of the kernel density estimate (step f, equivalent to the Surface Detection step b), except here we limit the kernel density estimate to photons below the calculated surface and use 30 photons rather than 75 photons, to better capture the lake bathymetry, which in general is more irregular than the
lake surface. Sub-surface ice layers near the edges of the lake are then reclassified as lake surfaces, thus sealing the bottom of the lake to the top at the lake edges (step g). In step h, we first perform a final smoothing, passing the resulting bottom photons through an iterative robust quadratic local regression (rloess) filter to remove outliers in the bottom estimates and then assign physical meaning to each photon (e.g. lake surface, bottom, surface ice, subsurface ice). The presence of surface ice is determined
based on the variability in thickness of the lake surface (i.e. a bimodal distribution indicates surface ice at some locations). We identify subsurface ice by the presence of weak returns above the lake bottom but below the surface (see Fig. 2 step b). The surface and bottom photons are then used to derive lake depth, where we apply a simple correction for refraction, described in Parrish et al. (2019), to produce a real corrected lake depth. Finally, in step i, we assign a final classification of a lake type using
properties of the local surface slope and the strength of the bottom return (Table 1). For example, a segment with a surface slope smaller than 0.03% as well as a distinct bottom (a photon density far exceeding the density of a background return) is given a lake classification of 'highly likely', whereas segments passing the same slope threshold but not showing a strong bottom return are identified as 'likely ice- covered' lakes. On the other hand, segments with a slope exceeding 0.3% and no significant
peak below the surface in the histogram are allocated to the 'highly unlikely' lake class.

| Class | Slope | Bottom reflection* | Lake probability | Lake characteristics |
|---|---|---|---|---|
| 1 | < 0.03% | strong | highly likely | very flat open lake, dense bottom photon returns |
| 2 | 0.3% < slope < 0.3% | strong | very likely | flat, open lake with presence of refrozen ice; dense bottom photon returns |
| 3 | > 0.3% | strong | likely | non-flat surface, possibly flowing water channel on sloping surface; dense bottom photon returns |
| 4 | < 0.03% | weak | very likely | very flat open lake; weak bottom photon returns |
| 5 | 0.3% < slope < 0.3% | weak | about as likely as not | flat open lake; weak bottom photon returns |
| 6 | > 0.3% | weak | unlikely | Non-flat surface; weak bottom photon returns |
| 7 | < 0.03% | none | likely | Very flat refrozen lake; no significant bottom photon returns |
| *: strength of the bottom reflection is defined by the ratio of the first two peaks in the 2 m interval histogram of non-surface photon heights within the segment. Strong bottom reflection: peak ratio > 3.5; weak bottom reflection: 2.5 < peak ratio < 3.5; no bottom reflection: peak ratio < 0 | | | | |

Table 1. Definitions for *Watta* lake classes

Two potential sources of ambiguity with the subsurface ice classification are: (a) the possibility for
specular returns and (b) apparent multiple surface returns which resulting from instrument echo. Specular returns over flat water (implying high energy return), return a strong surface as well as multiple layers below the surface spaced according to the ATLAS deadtime (0.45 m below the surface and a potential tertiary return below that). Echoes produced by electronic noise in the instrument, which also frequently occur very smooth water surfaces, can similarly produce a strong return at the surface

with double echoes at ~2.3 m and ~4.2 m below the surface (Martino et al., 2020; Lu et al., 2021). The categorization of subsurface ice (as in with Lake Ayşe in Fig. 3) are reliant on visual inspection. In this case, we assume subsurface ice because the layer shows trailing photons towards a weakly-resolved lake bottom rather than a distinctive sharp horizontal layer with no curved bottom return

## 3.2 Imagery Processing

A subsequent set of steps uses the lake depths extracted from the *Watta* algorithm to produce lake volumes from concurrent imagery, e.g. SkySat, PlanetScope, Sentinel-2 and Landsat OLI, requiring geolocation as well as the semi-automated identification of lake edges. Coregistration between ICESat-2 photon locations and imagery (Fig. 3 Step j) is managed by registering ICESat-2 elevations with the GIMP-2 DEM as an intermediary step (GIMP-2 is also used for georeferencing of Landsat), by
transforming the point cloud using the iterative closest point algorithm, which minimizes the square error between the two data sets (Besl and McKay, 1992). The point cloud from ICESat-2 is chosen to include a 0.2° latitude window surrounding the lake being resolved to include larger topography in the region (and thus avoid errors presented by ice motion). The large lakes used here are all located in strong topographic depressions (which are resolved in both the GIMP-2 DEM and ICESat-2) and can
therefore be assumed to remain relatively fixed.

To register imagery sources to one another, we standardize all imagery to the nearest Landsat image, using the arosics library in Python (Scheffler et al., 2017), which detects and corrects misregistrations of an input image (based on a reference image) at the sub-pixel scale. However, the coregistration of all other imagery sources to Landsat OLI first requires the delineation of lake
boundaries in order to exclude regions with moving surface water, which evolves rapidly and can be mistaken for fixed topography (which is more useful for geolocation). Here we calculated a normalized difference water index (NDWI) for each image, using a standard NDWI (with the green and NIR) bands to deliberately include regions with ice layers (as these are also detected by *Watta*), rather than the modified $NDWI_{ice}$ (which uses the red and blue band), per Yang and Smith (2013). Boundaries of lakes
(step l in Fig. 3) are calculated by using adaptive thresholding (Bradley et al., 2007) to generate a binary mask which is then used to identify individual water bodies. The use of adaptive thresholding avoids the limitations of any fixed NDWI threshold, especially relevant to PlanetScope data, which occasionally produces negative NDWI values. However, we note that this step has the potential to include partial ice layers (although visual inspection suggests that this was avoided with the test cases used here). To
coregister ICESat-2 to each imagery source, we also note that the ICESat-2 mission requirements list a geolocation accuracy of 6.5 m and a footprint size of ~11m (Magruder et al., 2021), which may potentially include multiple pixels of high-resolution imagery. To calculate a band value from imagery associated with a geolocated photon from ICESat-2, we find pixels in imagery within a 6 m radius and calculate a mean. In Step k, we calculate an empirical relationship between the depth estimate
calculated by *Watta* and a band value from coregistered imagery pixels. Finally, we use this empirical relationship to produce a depth estimate for the entire lake using eq. 1.

# Imagery Processing

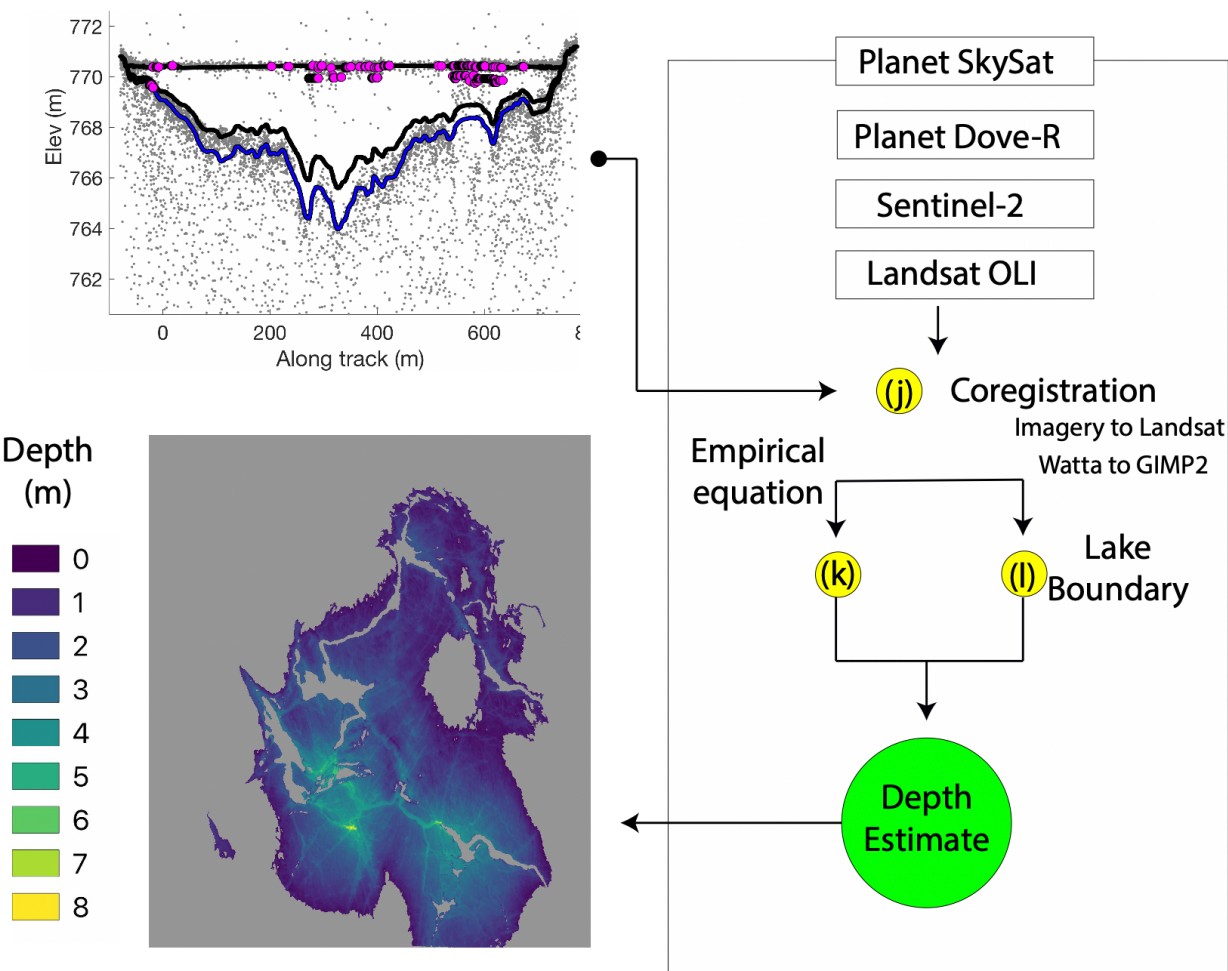

**Figure 3. Diagram of imagery processing steps accepting *Watta* outputs as input (top left) and producing lake depth estimates (bottom left). *Watta* lake depth profile shown is for Lake Ayşe using RGT 841 on May 23rd and Planet SkySat imagery on May 22nd.**

## 4 Evaluating Methodology

### 4.1 Physical constraints of the test dataset

The test dataset provides a diversity of lake types, with the largest surface area calculated at 5.6 km$^2$ and a maximum *Watta*- calculated corrected depth at 10.3 m. A number of lakes contain substantial ice cover. In the following sections, we discuss several lakes in greater detail which present a diverse set of

conditions with which to evaluate both the *Watta* algorithm and the imagery sources used to extrapolate lake volumes. Locations of the lakes can be found in Fig. 1.

320       Lake Ayşe was selected for closer examination because despite the dense photon cloud, its relief in surface and bottom, combined with the presence of ice cover, pose a challenge for our detection algorithm. Additionally, multiple imagery sources were available within a 5-day window at this location. Lake Zadie is chosen because it represents an ideal case for the algorithm, while Lake Cecily is chosen because two beams passed over the same lake with SkySat imagery available one day afterwards. A basic assumption we make in this study is that the lake bottom remains relatively

consistent over several days, although past research on two lakes in Western Greenland has estimated lake bottom ablation rates at 6.5 cm/day on the bottom of the pond (Tedesco et al., 2012). We assume that this is the primary physical source of uncertainty in the empirical calculation, as the relationship will degrade with temporal distance from the ICESat-2 pass. However, where changes in the bathymetry are not uniform, we can potentially make inferences about drainage mechanisms (e.g. the

forming and deepening of crevasses). Cross-sections of all lakes used for development showing the lake top, bottom, and bottom value corrected for refraction as calculated by *Watta*, along with lake top and bottom as calculated empirically from imagery, are shown in Supplemental Fig. S4 and their coordinates and relevant statistics listed in Table S1.

**4.2 Physical constraints of the test dataset**

The most rigorous weighting system used for the algorithm, using only lake classes 1, 2, 4 and 7 (see Table 1), succeeded in automatically detecting 49 out of the 50 lakes identified in the available imagery, with two likely false positives (i.e. not confirmed by NDWI values exceeding 0.2 in the imagery). A less rigorous weighting system, including lake class 3, detected the 50th lake, but resulted in a large

number of false positives in areas of steep, and rough topography, where abrupt changes in the photon elevations are misinterpreted as bottom reflections by the algorithm.

      In the absence of simultaneous *in situ* data, we evaluate the performance of the algorithm based on visual inspection (comparing ATL03 heights with *Watta* calculated depths as shown in Fig. S3). Additionally, where an empirical relationship with imagery is successful (a high correlation coefficient

value between imagery-derived depths and *Watta*-derived depths,), we take this consistency for partial evidence that ICESat-2 and imagery sources have detected bathymetry correctly, although we note that this metric is only applicable to the specific lake, not all the lakes in the region. The most successful bottom retrieval occurred where ice cover was minimal, the density of photons was high and where the bottom slope was relatively uniform (e.g. Lake Zadie). The presence of ice near the surface (between

the surface and 1 m below the surface) frequently obscured lake bottom detection (e.g. reference ground track (RGT) 1222, Lake 3 in Fig. S3), although in some cases only partially; however, the presence of subsurface ice did not always preclude the presence of a strong bottom return (e.g. Lake 7, RGT 1169, Fig. S3). The algorithm therefore indicates the presence of surface/near- surface ice, but does not automate the removal of the calculated bottom return due to ambiguity. We can confirm the presence of

an ice layer both by visual inspection of the imagery and by comparing standardized NDWI values calculated from imagery coincident with the ICESat-2 track (Fig S1a). We note that for at least one

case, (RGT 1108 Lake 6, Fig. S3), the designation of "lake" was ambiguous, as this could be treated as either a shallow lake containing a large amount of subsurface ice, or as a slush layer (a number of which were identified elsewhere).

**4.3 Evaluating Data Sources for Imagery-based Depths**

Total uncertainty for the empirically-based depth estimates from imagery is comprised of uncertainty in ICESat-2 geolocation, uncertainty from the *Watta* algorithm itself (which operates at a vertical resolution of 0.1 m), from the resolution of the imagery, from the uncertainty in alpha coefficients calculated from the empirical method and finally from physical changes in the lake occurring between
the time that imagery is captured and the ICESat-2 pass. The empirical calculation is less likely to be affected by physical changes in the lake when the lake surfaces calculated by imagery vs altimetry differ by less than a meter; here we estimate precision with a simple $R^2$ value.

Past work has considered either the red or green band for developing depth estimates (Moussavi et al., 2020; Williamson et al., 2018), though *in situ* validation was limited at the time (Pope et al.,
2016). Figure 4 compares $R^2$ values from empirical estimates derived from the red vs green band for lakes classified according to the maximum lake depth calculated by *Watta*. In agreement with Moussavi et al. (2016), for Landsat 8 (Fig. 4d), Sentinel-2 (Fig. 4c) and SkySat (Fig. 4a), the empirical depth estimates for the red band showed higher fidelity with *Watta*-based depths for shallow lakes, while the green band showed greater fidelity for deeper lakes. Of six lakes where a maximum depth exceeds 7.2
m and where the imagery source is Landsat 8, two lakes (RGT 1222, Lake 8, 12 in Supplemental Profiles) show both red and green-band based profiles being unable to resolve the deepest points in the lake. For two additional cases (RGT 1222, Fig. 4e, Lake Zadie, Lake 14, 17 in Supplemental Profiles), the green band was able to resolve very deep lake depths, while the red band was not. This implies that for SkySat, Sentinel-2 and Landsat 8, the green band is able to resolve bathymetry at greater depths and
emphasize cracks at the bottom of the lakes. The major exception is PlanetScope data (Fig. 4b), where the red band consistently showed greater fidelity to *Watta*-based estimates while green band estimates produced unrealistic depth estimates, although there were a limited number of lakes where coincident PlanetScope imagery was available. We note that because this method is empirical, future users would be able to select bands or combinations, as with the average of the panchromatic and red band used by
Pope et al. (2016).

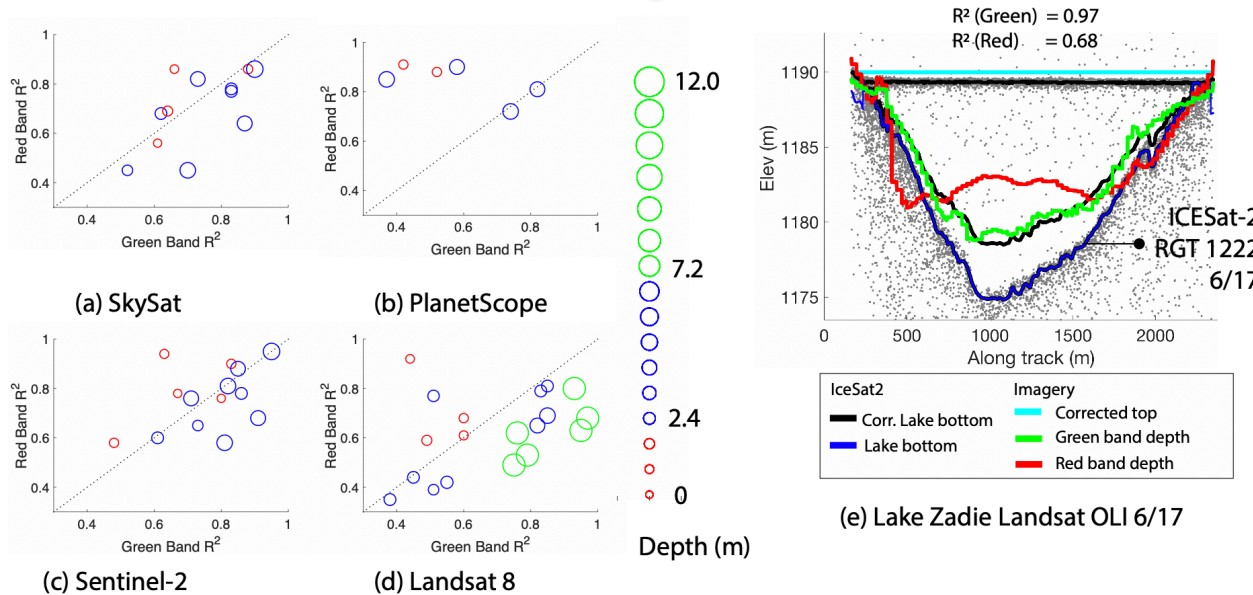

**Figure 4. Comparing R2 values from empirical estimates calculated with the red band vs the green band from multiple imagery sources, with lakes classified by maximum lake depths as calculated by *Watta*. (a) Planet SkySat TOA reflectance (b) Planet PlanetScope surface reflectance (c) Sentinel-2 TOA reflectance (d) Landsat 8 OLI TOA reflectance (e) *Watta*-calculated and imagery-derived depths: Lake Zadie based on Landsat OLI.**

To demonstrate the robustness of the *Watta* algorithm, the impact of band choice, and the sensitivity to absolute lake depth, we show depths calculated from two beams passing over Lake Cecily on June 13[th], followed by retrieval of SkySat imagery on June 14[th] and Sentinel-2 on June 16[th] (Fig. 5 a,b and RGT 1169 Lake 5(6) in Supplemental Profiles). Over this spot, covered by the 3l beam on this pass, the green band shows higher $R^2$ values for both Sentinel-2 and SkySat, but lower $R^2$ values for the red band. This is consistent with the greater depths calculated from the 3l beam, which approach the 6 m depth at where the performance of the green band is expected to improve. The use of the green band in both the 3l and 3r cases allows for finer bathymetric relief to be captured in both SkySat and Sentinel-based depth estimates, with the finer resolution of SkySat capturing substantially greater detail (Fig. 5c, box). We note that even when high $R^2$ values are calculated between the empirical estimate and *Watta*-calculated depths, unrealistic depths can result when lakes drain or fill rapidly, and low-resolution imagery can potentially resolve the height of a lake surface inaccurately (Fig. S2).

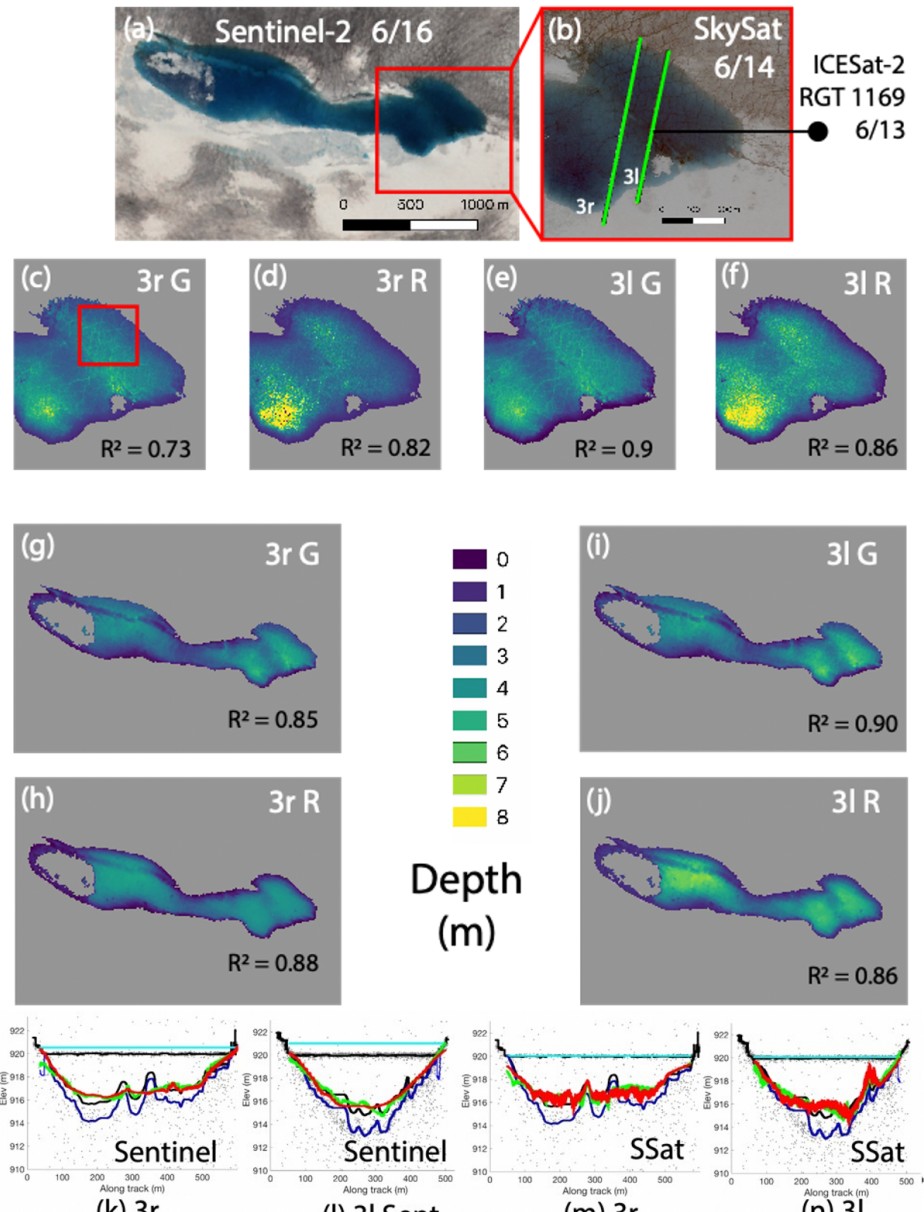


**Figure 5.** *Watta*-**calculated and imagery-derived depths: Lake Cecily based on Sentinel-2 (k,l) and Planet SkySat (m,n); Lake Cecily false-color imagery from Sentinel-2 (a) and SkySat with ICESat-2 beam 3r and 3l (b); Imagery-derived depths from Planet SkySat (c-f) and Sentinel-2 (g-j) with band/beam combination as shown. Estimates from the green band are indicated by "G" while those calculated from the red band are indicated by "R". is Red box in (c)**

**highlights region where underlying crevassing is captured**

Within Figure 6, we show the depth evolution of Lake Ayşe over 5 days, both along the ICESat-2 ATL03/*Watta*-calculated profile and the lake volume estimates then constructed from imagery using the empirical equation. Increasing lake volume is demonstrated both by the expansion of the surface

area of the lake through time (right column) and the rise in the lake level (cyan line, left column). Planet SkySat estimates at a 1 m resolution (Fig. 6c) show the greatest level of detail of crevassing at the bottom of the lake although PlanetScope estimates, at a 3 m resolution (Fig. 6d,e) are comparable. We note that PlanetScope data showed variations in the fidelity to *Watta*-based estimates between the green band vs the red band depending on the instrument. The bottom relief is maintained at depth with the

PlanetScope resolution (3 m), and future work could generate more reliable depth estimates by calibrating empirically-based depths to Sentinel estimates, which could provide very high-resolution depth estimates while also leveraging the high temporal frequency of PlanetScope data collection.

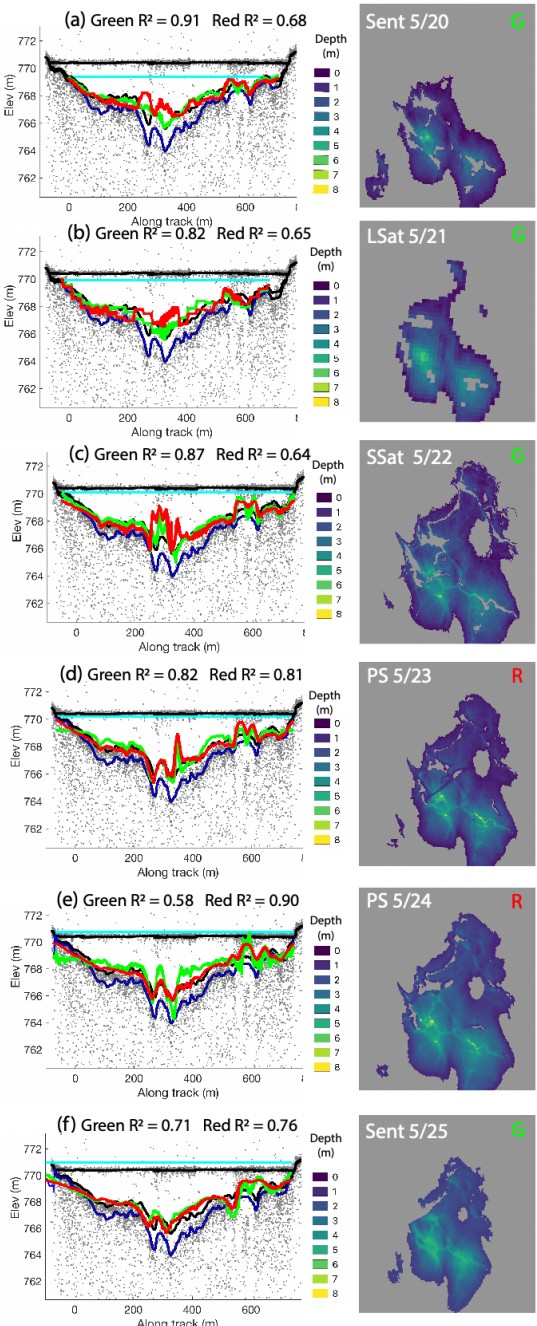

**Figure 6. Lake Ayşe, filling over five days between May 20th and May 25th, with ICESat-2 pass on May 23rd. Left column: Profiles with *Watta*-calculated and imagery-derived depths from the green, indicated by "G" and red bands, indicated by "R" (with corresponding R² values inset), legend same as Fig.4a. Right column: Depth values derived from empirical estimate, with imagery source , date collected, band used for depth estimate shown. Imagery source abbreviations are as follows: "LSat" for Landsat, "SSat" for Planet SkySat, "PS" for Planet PlanetScope and "Sent" for Sentinel-2.  Path of ICESat-2 spot over Lake Ayşe is shown in Fig. 1**


**5 Capturing Lake Drainage over the Melt Season**

Our improved ability to track lake depth and volumes using the combination of ICESat-2 and multi-sensor imagery can potentially provide new insights into patterns of lake drainage. Lake Julian (Fig. 7a) was selected for closer examination because on May 15[th], both the airborne Operation IceBridge
mission and ICESat-2 passed over this region, providing a unique stack of both airborne and satellite data. While we show only very high-resolution Operation IceBridge CAMBOT imagery here, other instruments aboard OIB could potentially provide valuable insight into the state of both surface hydrology and firn characteristics in future work. Additionally, there are cloud-free ICESat-2 RGT 727 passes over this lake both on May 15[th], 2019 and on August 14[th], 2019, providing a profile of the lake
both when it was filled as well as after drainage. A second lake, Lake Niels (Fig. 7a) is examined briefly primarily to provide context. Although no altimetry estimates are available over Lake Niels, imagery sources reveal a very different evolution and drainage pattern despite its being located only 3500 meters from Lake Julian, and consequently subject to many of the same atmospheric drivers. Within the larger region shown in Fig. 7a (see also Supplemental Fig. S4), the percentage of the ice sheet surface covered
in liquid water, as measured by the percentage of the region where NDWIice values exceed 0.2, remains constant at around 3% from May through June, with meltwater being spread more uniformly (less visibly collected in large hydrological features) over the ice sheet early in the season and shifting to larger lakes later in the season (Supplemental Fig. S4). We note that this measure of melt extent does not translate directly to consistent meltwater volume; meltwater underneath the snowcover on Lake
Niels was not estimated earlier in the season, while the deeper lakes which are present later in the season will contain larger water volumes.

Although elevation decreases overall toward the northwest, both lakes coincide with large-scale surface depressions calculated from the GIMP-2 DEM (Fig. 7) and this region experiences comparatively low ice velocities. Lake Niels is located in a deep surface depression whereas the
corresponding surface depression for Lake Julian is relatively shallow. Because both imagery and ICESat-2 are coregistered to the GIMP-2 DEM, we presume that all inaccuracies will be consistent (i.e. even if geolocation is incorrect in absolute terms, imagery and ICESat-2 should overlap).

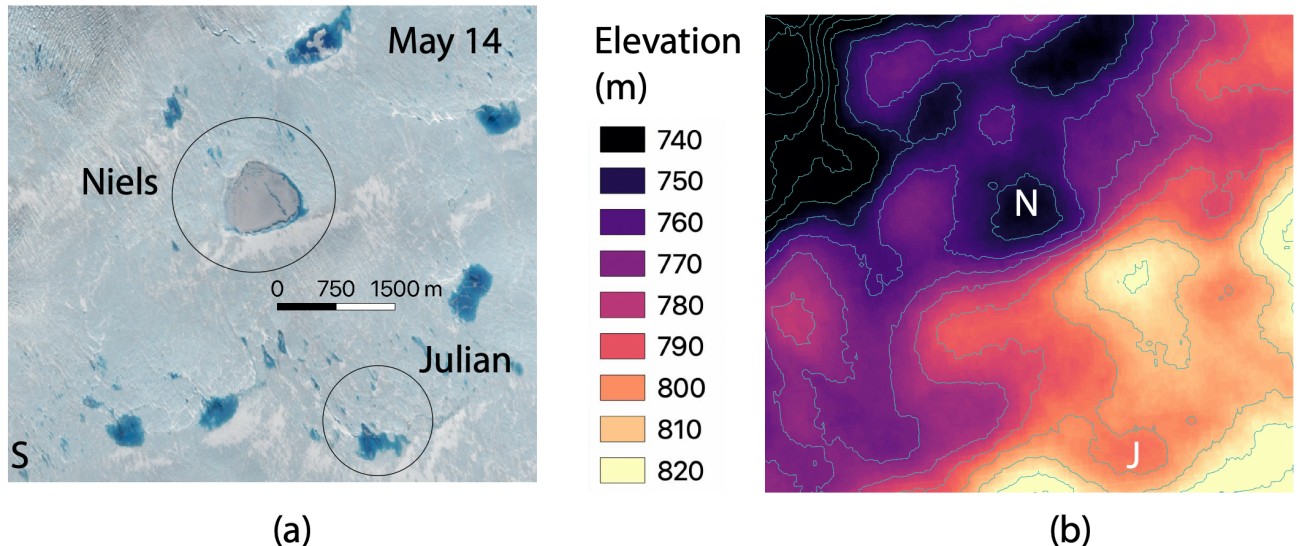

Figure 7. (a) Lake Niels and Lake Julian, shown on May 14th, 2019 using Sentinel-2 imagery as described in the Data
section. (b) GIMP-2 DEMP shown at same location with contours shown at 10 m intervals.

## 5.1 Drainage Mechanisms over Lake Julian from airborne and satellite-based imagery

The volume of Lake Julian begins to increase substantially on May 9th and reaches a maximum volume
between May 25th and May 29th (Fig. 8j-l). After June 1st, the lake begins to lose volume until only
remnants are present on June 10th, which disappear almost entirely by June 19th (Fig. 8m-o). The
surrounding region (i.e. a kilometer to the north and west) contain smaller bodies of water connected by
streams. We note that while larger streams can be captured by Sentinel-2 imagery (Fig. 8j), many of the
smaller streams present later in the season can only be reliably detected with imagery at a 1 meter
resolution or below (e.g. Fig. 8r, Supplemental Fig. S5). The progression shown in Fig. 8 captures the
development of an efficient drainage system over the season.

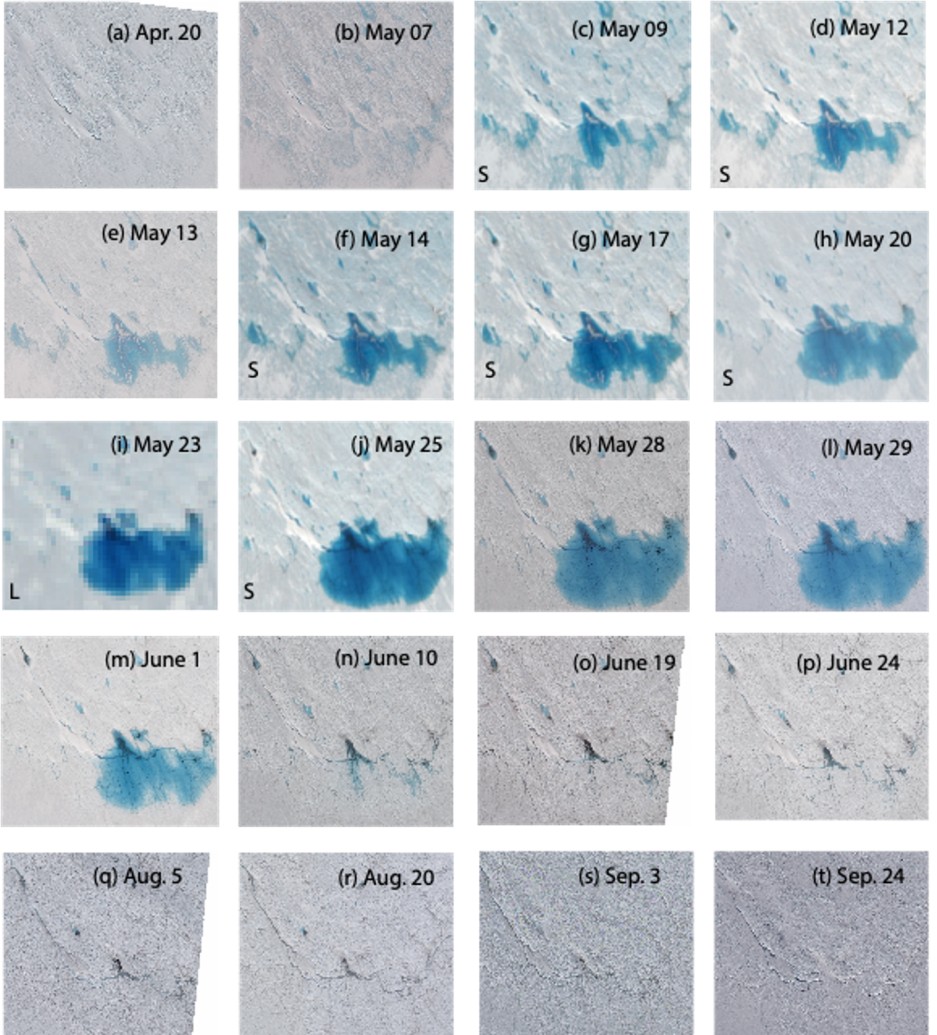

**Figure 8. Imagery over Lake Julian shown from April 20th, 2019 through Sep 24, 2019. All images use Planet SkySat Visual data unless otherwise indicated by letter on bottom left, with S indicating Sentinel and L indicating Landsat**

Three potential drainage mechanisms can be observed over lake Julian. Firstly, we note a small stream ending in a spray of snow (alternatively, an ice bridge) which is potentially indicative of a moulin (Fig. 9b). However, an overflight of Operation IceBridge on May 15th (Fig. 9a) does not definitively show a moulin at the end of the incision, allowing for the possibility that the actual drainage occurred under the overlapping ice bridge. We also identify the point labelled B in Fig. 9a,b (also in

Fig. 10d,e) as another potential drainage point. Presuming that drainage occurs at either location, lake volume could still increase slowly (as it does between May 9th and May 25th) if the inflow rate exceeds the outflow rate of the lake. This dynamic is captured in a previous *in situ* study of lake drainage, which indicated that drainage through a moulin decelerated as the hydraulic head between the lake and the moulin declined (Tedesco et al., 2013).

485        However, following late May, SkySat imagery captures the development of a second stream directly south of the initial potential moulin (Fig. 9b). This small stream, which flows downstream (Fig. 7b) deepens throughout the season (Fig. 9c) with a very deep incision shown distinctly in imagery shown on September 24th (Fig. 8t). The development suggests that the relatively slow initial drainage from the potential moulin or Point B accelerated due to increased drainage from a second small stream,
draining Lake Julian almost entirely between June 1st and June 10th (Fig. 8m-n). While smaller pre-existent streams to the right (Fig. 9b) may also have facilitated drainage, we assume that the newly incised stream to the left is the most likely cause for the rapid drainage due to the timing of its appearance. We note that smaller bodies of water are still apparent on the surface after the drainage of Lake Julian, some connected by very small stream networks, which appear to be frozen-over by
September 24th (Fig. 8t).

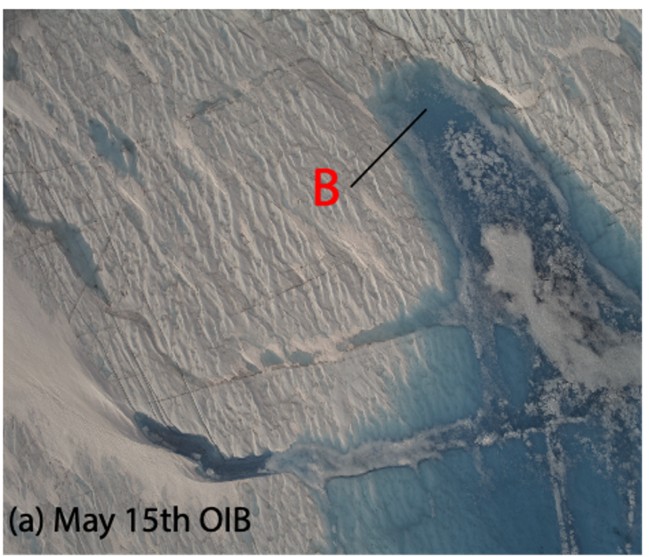 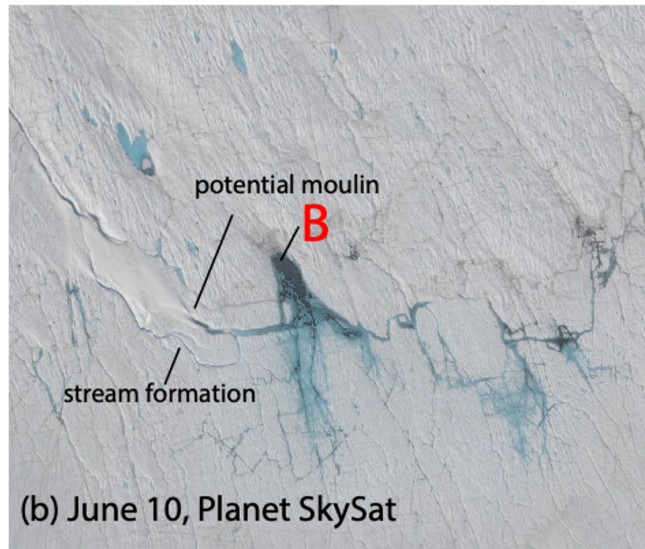

**Figure 9. Mechanisms of lake drainage over Lake Julian. (a) CAMBOT imagery from Operation IceBridge flight on May 15th, 2019 (30cm resolution). (b) SkySat imagery (~1 m resolution) on June 10th after large-scale drainage**


**5.2 Drainage Mechanisms over Lake Julian Captured using *Watta***

Lake Julian reached a volume of 268120 m$^3$ on May 14th, which was calculated using the green band from Sentinel-2 imagery on May 14th in conjunction with a *Watta*-based depth calculation from an ICESat-2 pass on May 15th (Fig 10a,b). The time lag introduces uncertainty due to possible lake
ablation. We assume that this uncertainty is not due to the discrepancy in dates (imagery having been captured on May 14th whereas a deeper lake depth was captured by ICESat-2 on May15th). This is because the methodology accounts for changing lake depths, presuming minimal lake ablation, matching *Watta*-calculated depth to the surface height where the edge of the lake is indicated by imagery (see Methods). The depth values calculated for May 14th indicate a relatively shallow lake (less
than a maximum 4 meter depth).

A comparison between the two passes of ICESat-2 RGT 727 (the second on August 14[th]) indicate uneven lowering in this region and potential slight ice motion, e.g. a small southward shift in the lowest point of the depth profile (pt B) calculated on May 15[th] vs on August 14[th] (Fig. 10c). We estimate large-scale surface lowering around ~1 m, based on the lowering calculated at higher elevations (Fig. 10c,d, where the x axis, showing the distance from an arbitrary start point, is greater than 800 m). By contrast, elevation changes where surface hydrology features exist show enhanced incision of a pre-existent stream/drainage point as well as the development of a new stream (Fig. 10 c,d, x-axis value between 0 and 200 m). The deepening of the lowest point in the lake could be the product of ice motion, but we assume that the elevation change of 2-3 meters at this location is the effect of lake ablation. This is due to the locations of ice layers being well-matched between imagery and *Watta*-calculated features (discussed shortly), suggesting that any ice motion was adjusted for in the geolocation step.

In addition to locations where *Watta* calculates a lake surface (Fig. 10d and Fig. 10e, label "B"), *Watta* also identifies regions where ice cover is probable. These are shown in cyan in Fig. 10d at locations A, C, D and in imagery in Fig. 10e. Whereas lake surfaces are calculated at the horizontal resolution of the ICESat-2 ATL03 photon cloud, the ice surface class is assigned at a coarser resolution. This is because the Interpretive module assigns the "ice surface" class based on the presence of a flat surface under an overlying layer with more varied topography. While currently, the algorithm potentially overestimates the extent of these regions, a first automatic pass can be used to identify larger regions where ice surfaces exist, after which manual inspection can then identify specific ice layers. Shown in Fig. 10d are the lake (B) as well as three additional points where we identify ice layers using both *Watta* and manual inspection (A, C, D). Point B is also captured in Operation Ice Bridge CAMBOT imagery on May 15[th] (Fig. 9a). This is potentially a drainage point which retains meltwater as late as August 14[th]. This location is also covered by a floating ice layer on May 14[th], suggesting that an ice layer had formed at the same place and settled at this point following drainage in the previous season. Point C corresponds to a deeply incised stream (which is not captured in the *Watta* profile calculated on May 15[th]) while Point D corresponds to a smaller stream; both of these points were covered by Lake Julian during the last half of May. We note that the designation of Point A is more ambiguous as it is collocated with to an incision which was previously a stream, but is weakly-resolved in both *Watta*-based estimates and in the SkySat imagery collected on August 20[th]. The Watta designation of "ice surface" here is likely, but not unambiguous, as this method will capture both real ice surface and false dual returns as detailed in Section 3.1. The main attributes typical of the false dual returns are a strong top surface over surface water in a flat region, followed by weaker returns at predictable intervals (~0.45 m and possible ~0.90 m for the specular return, ~2.3 and ~4.2 m for the instrument echo). In this case, a specular return would be the most likely cause for a false dual-return due the spacing. However, we first note that the surface in this region is not flat and we do not see the predicted strong surface return followed by a weaker echo ("surface" and "ice" layers are of equal thickness). Additionally (a) in the case of point B, the top layer contains no dual return (b) in the case of C, a distinct gap occurs in the surface. Both of these correspond to ice/water in imagery. For point A and point D, imagery suggests that these points occur at a convergence of streams. These could, however be either water or ice as no distinctive bottom return is detected. With the current available

information, the "ice surface" detection will still require manual inspection; future improvements to the code may account for the known issue with false dual returns as knowledge in this area develops.

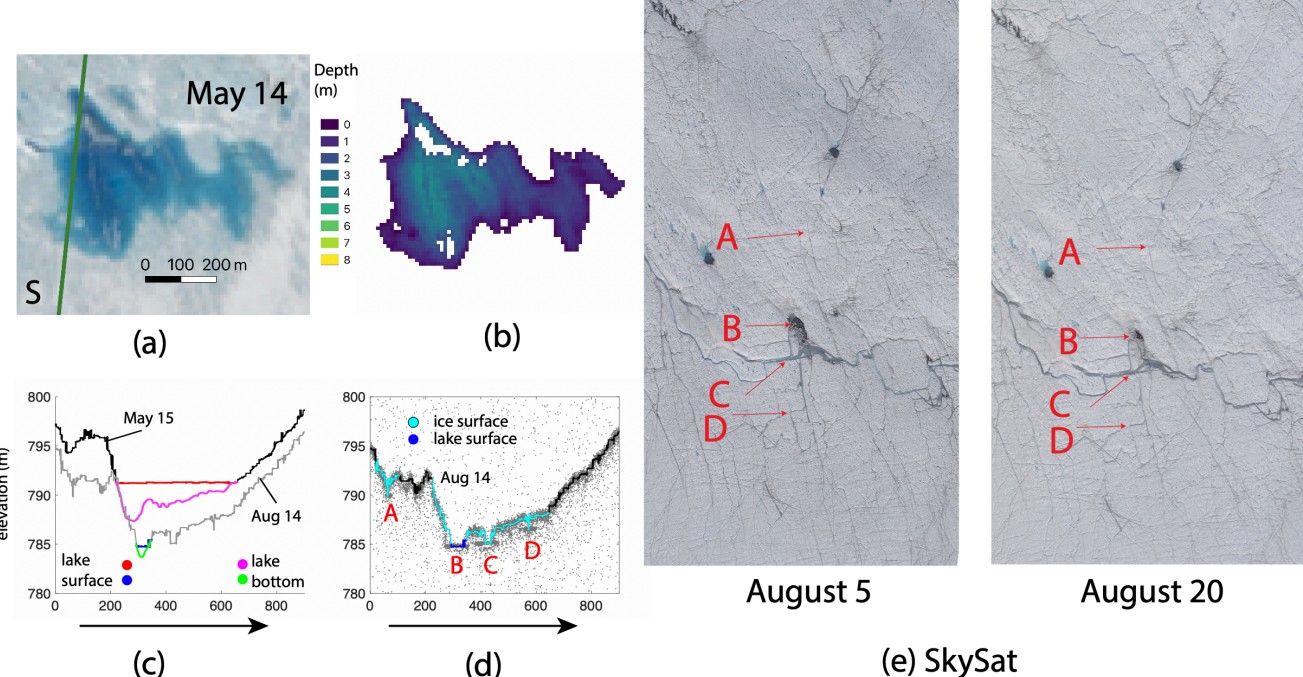

Figure 10. ICESat-2 RGT 727 over Lake Julian. (a) Sentinel-2 based imagery acquired on May 14[th] 2019 with ICESat-2 RGT 727 gt1l (occurring on May 15th, 2019 and August 14th, 2019) overlapping in green. Operation IceBridge overpass on May 15th, 2019 is directly coincident with the ICESat-2 line. (b) Lake depth derived from *Watta* (Panel c) and Sentinel-2 imagery (Panel a) based depth. (c) *Watta* calculated from ICESat-2 on May 15th and Aug 14th. (d) *Watta*-calculated surface features over photon cloud on Aug 14th . Points A-D discussed in text (e) Points A-D shown over Planet SkySat imagery collected on August 5[th] and August 20th

### 5.3 Lake Niels, Partial Drainage and Refreeze

In comparison to Lake Julian, Lake Niels begins and ends the melt season as a frozen lake. By May 9[th], the ice surface begins to melt and the lake surface area expands substantially. However, by May 13[th] (Fig 11d), imagery captures an insulating layer of snow, after which the lake expands into June 26[th]. By July 20[th], slow lake drainage is evident via a stream which is identified in Fig. 11a and is first observed to contain a substantial quantity of liquid water on June 19[th] (Fig. 11h). We observe that the stream is clearly incised both on April 20[th] (Fig. 11b) and on September 24[th] (Fig. 11m), when the lake is likely frozen over, based on imagery. In comparison to Lake Julian, which was located in a relatively shallow depression, Lake Niels is located in a deep depression (Fig. 7) and neither drains early in the season nor connects to an efficient drainage system. Although these lakes are subject to similar atmospheric drivers, the differences in drainage patterns highlight how local topography and the corresponding depth of lakes can influence how meltwater is either retained on the ice sheet vs drained downstream or into englacial or subglacial pathways. We note that surface topography, in turn, is

strongly influenced by basal topography, thus linking total runoff to surface expression of bed
topography (Ignéczi et al., 2018).

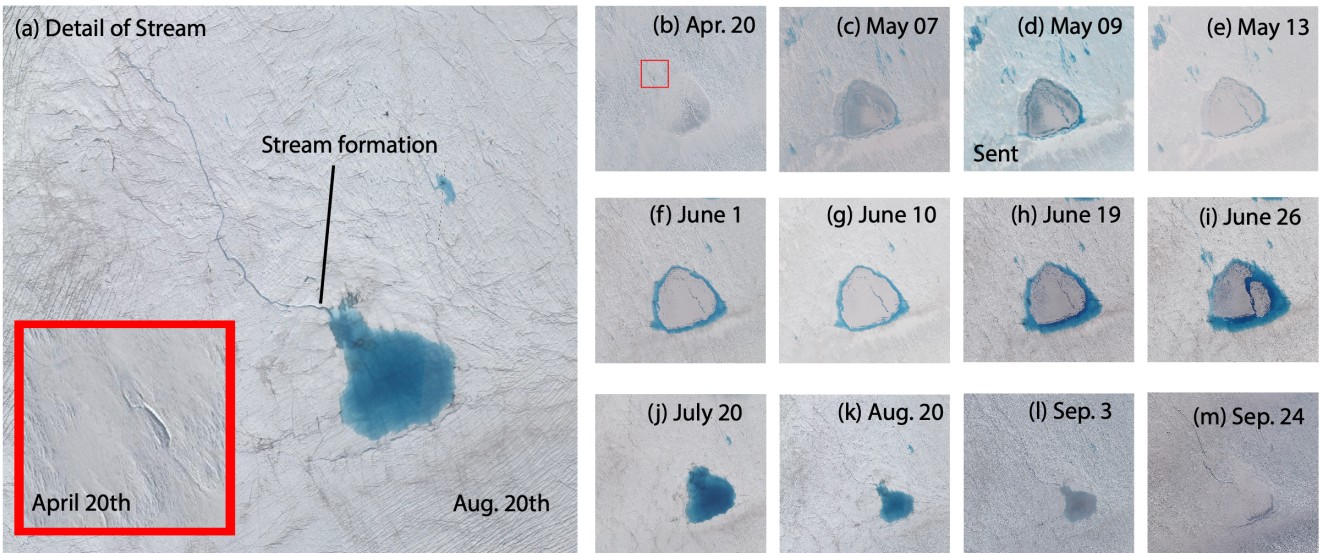

**Figure 11. Lake Niels, shown between April 20th and September 4th, 2019. (a) shows stream detail from Aug. 20 (main panel) with stream detail on April 20<sup>th</sup> (inset, with region indicated in Panel b). All imagery is from Planet SkySat Visual imagery except on May 9th (where Sentinel-2 imagery is used, indicated with "Sent").**

**6 Conclusions**

This study represents initial work developing the *Watta* algorithm for lake depth estimates as well as subsurface ice detection, using a unique stacked dataset over Western Greenland during the intense 2019 melt season. We demonstrate the potential of ICESat-2 for automated lake detection and depth estimation, as well as how empirically-derived depths derived from a combination of imagery sources
can complement each source's strengths and weaknesses. For example, while Landsat is only available at a low resolution, it provides a rich historical record as well as high geolocational accuracy (at 5 m), which is leveraged here to better geolocate imagery from Planet Labs. Similarly, while PlanetScope data contains several known issues with radiometry and geolocation, imagery is available at a high spatial and temporal resolution. As demonstrated in our test cases, a time series constructed from
multiple sources can provide valuable information about the evolution of ice cover and drainage mechanisms in addition to volume estimates. Given the accelerating sophistication of altimetry-based observations, ongoing efforts to improve geolocation, radiometric quality or temporal frequency of high-resolution imagery are crucial. Additionally, the availability of simultaneous imagery and altimetry would enhance the capabilities of other satellite imagery sources to fill out the time series by providing
a calibration standard.

      While this initial study focused on lakes in grounded ice in Greenland, *Watta* can potentially be applied to Antarctic melt lakes as well. Additionally, lake depths calculated empirically can potentially

be used to calibrate physically-based methods towards developing ice-sheet-wide timeseries for the evolution of surface hydrology.

Our algorithm successfully detects a wide variety of lake types automatically, and can be applied to the growing set of ICESat-2 and imagery data over large sections of Antarctica and Greenland. Identification of narrow stream features on sloping surfaces, however, still needs visual verification due to a large number of false positives. This will be addressed in future work, together with adding features to the interpretive layer, including slush layers as well as cracks, using the Planet SkySat imagery

dataset for testing purposes. In addition to these improvements, *Watta*, which is currently written in MATLAB, is available in github with documentation, but will eventually be moved to an open-source language.

**Author contribution**

Both authors contributed equally to this work.

**Acknowledgements**

We would like to acknowledge Lauren Andrews for her help in identifying drainage patterns, Jeremy Harbeck for his assistance with Operation IceBridge CAMBOT imagery and the NASA Small Satellite Databuy Pilot Program as well as Planet Labs for the use of Planet Imagery. B.W. acknowledges

funding by NWO grants 016.Vidi.171.063 and OCENW.GROOT.2019.091. R.T.D. was funded by the ICESat-2 Project Science Office.

**Competing Interests**

Bert Wouters is editor of The Cryosphere.

**Data Availability**

Code for Watta can be found here: https://github.com/kettleofmonkeys/Watta.git
Imagery, IDs for which can be found in supplemental information can be accessed at the following urls:
For Planet Imagery: https://www.planet.com/
For Sentinel-2 and Landsat 7 and 8 imagery: https://earthexplorer.usgs.gov/

For ICESat-2 ATL03: https://nsidc.org/data/atl03

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
