# Peer review of "Supraglacial lake bathymetry automatically derived from ICESat-2 constraining lake depth estimates from multi-source satellite imagery"

_The Cryosphere, 2021_

## Referee Comment (RC1)

**Supraglacial lake bathymetry automatically derived from ICESat-2 constraining lake depth estimates from multi-source satellite imagery**

Tri Datta and Bert Wouters

Jennifer Arthur (Referee)
jennifer.arthur@durham.ac.uk

**General comments:**

This manuscript presents a new algorithm, 'Watta', for automatically extracting supraglacial lake depth estimates using ICESat-2 geolocated photon heights (ATL03) which are then used to validate empirically-derived lake depths from Landsat 8, Sentinel-2, Planet Labs Skysat and PlanetScope imagery. The authors test the algorithm performance on 46 supraglacial lakes near Jakobshavn Glacier in West Greenland during an intense melt season (2019). Finally, they use this stacked dataset in combination with Operation IceBridge imagery to track volume, drainage mechanisms and ice cover evolution of two individual lakes in this region.

Supraglacial lakes form in the ablation zones of Greenland and Antarctica during the summer melt season and can impact ice sheet dynamics, making lake detection and depth retrieval important. However, lake volumes have been difficult to quantify due to a lack of *in-situ* measurements and uncertainties associated with image-based methods. This manuscript builds upon other recent studies by applying a novel method for lake depth extraction, which is the first application of high-resolution Planet Labs satellite imagery to calculate supraglacial lake depths in combination with other imagery sources and ICESat-2 heights. It also provides useful insight into lake dynamics and ice cover evolution, which to date have been limited by the comparatively coarse resolution of publicly-available satellite datasets (Sentinel, Landsat).

Therefore, it is my view that the findings are of broad interest to the cryospheric community and represent a promising step forward for studying supraglacial hydrology and dynamics. I look forward to seeing further development of this method and its applications elsewhere on the Greenland and Antarctic ice sheets, particularly on floating ice shelves.

In general, this is a well-written manuscript and most of my comments are relatively minor. I would like to see in places some additional detail around the discussion of lake depth retrieval methods (see specific comments).

Lastly, The Cryosphere's data policy states that "Authors are required to provide a statement on how their underlying research data can be accessed. This must be placed as the section "Data availability" at the end of the manuscript." Although the authors state at the end of the manuscript that the Matlab code will be converted and shared publicly, I would like to see this section added including statements of how Landsat, Sentinel and Planet imagery can be accessed.

Once the authors address these points and my comments below, I can therefore recommend that this manuscript is suitable for publication in *The Cryosphere*.

**Specific comments:**

Line 17: Specify Landsat 8 OLI, as you are specifying Sentinel-2.

Line 38: 'a common feature on large parts of the ice sheets' – this should be ice sheet (singular) as the paragraph has only discussed Greenland so far.

Line 40: ' (…) of both ice sheets'. So far, Antarctica has not been mentioned in the text. Perhaps add an additional 1-2 sentences in the previous paragraph introducing meltwater production and Antarctic-wide surface meltwater and supraglacial lakes.

Line 43: Suggest rewording to: 'The complex links between (…) can potentially be deduced (…)'.

Line 45: Specify here that feature types are 'supraglacial'.

Line 46-7: I think detail could be added here outlining both the physically-based and empirically-based methods, together with a slight clarification of the wording, as the physically-based approach has been applied to other optical imagery such as Aster (Sneed and Hamilton, 2007), Landsat (Banwell et al., 2014) and Sentinel. It should also be added to the text that the physically-based method assumes a minimal impact of wind-driven surface waves, low slopes of lake bottoms and a homogeneous lake-bottom albedo on lake depths, which may be particularly important in Greenland (Sneed and Hamilton, 2011). I believe the empirically-based method was first applied to WorldView2 by Legleigter et al. (2014), so I suggest citing this work too (see below for full reference).

Line 49: Can you include a specific lake depth limit here? I believe Pope et al. (2016) found that the maximum lake depth that could be derived from the empirical depth retrieval method using *in-situ* estimates was 5 m.

Line 55: Specify resolution here (e.g. ≤ 3 m, daily) to demonstrate improvements over Sentinel-2.

Line 61: Could you specify by how much the physically-based estimates tended to underestimate lake depth?

Line 65: Specify native resolution (0.7 m).

Line 68-70: This sentence is slightly hard to follow – adding 'from' before 'multiple imagery sources' may improve the clarity?

Line 72: It is not clear to me what the representative sections are that are referred to in this sentence – is this part needed?

Line 75: Can you cite any work here to support that this was an unusually intense melt season, e.g. Tedesco and Fettweis (2020)? I would also specify 2019 here too, just to clarify to the reader.

Line 78: I think this is the first place CAMBOT is used as an acronym, so include the full name (Continuous Airborne Mapping By Optical Translator).

Line 85 (Figure 1): Please make the scale bar in the top left panel bigger, and add lat-lon labels to the main panel. Perhaps also add '(A)' to the Lake Ayse label to make clearer how the Skysat image relates to the main panel. In the Figure caption, specify that RGT = repeat ground track. Finally, a small comment but maybe call SkySat boxes 'grey' rather than white in the caption so that it is clear when looking at the main panel.

Line 100: Specify ICESat-2 confidence levels (i.e. low, medium, high) in brackets.

Line 116: Are there any studies you can cite here to demonstrate the PlanetScope radiometry issues? Possibly Saunier (2020)?

Line 121: What is the vertical accuracy of the GIMP-2 DEM? Also, include the GIMP-2 dataset citation here (Howat et al.).

Line 143: Specify how lake boundaries are delineated? Discussion of NDWI does not come until Section 3.2, consider moving some of this into the Methods section.

Line 195: I wonder if you have considered what the effect is of wind-driven scattering on lake surface roughness and on the surface photon return?

Line 218: Specify that NDWI$_{ice}$ uses the blue and red bands.

Line 220: Is 'limitations' a better word here, rather than 'limits'?

Line 246: I think Banwell et al. (2019) calculate a lake-bottom ablation rate of 20.3 mm day$^{-1}$ on McMurdo Ice Shelf. Perhaps worth adding in?

Line 315: Be explicit in the Figure 5 caption that G = green and R = red in panels c-j.

Line 330: In Fig. 6 caption, write abbreviations in full again for clarity (

Line 345: 'Liquid water (…) remains constant at around 3%' – over what period? Looking at Figure S4, the surface water extent appears to increase?

Line 346: Can you refer to a figure/results here to support this statement?

Line 355: Could you add lake outlines to Panel b of Figure 7?

Lines 360-62: Refer to Fig. 8(j-l) end of first sentence, and (m-o) in the next sentence.

Line 383: Refer to Fig. 8m-n at the end of this sentence.

Line 390: Looking at the SkySat image in Figure 9b, incised streams are also visible to the right of the image; could Lake Julian not also have drained through these?

Line 401: Should this refer to Fig. 10c here, not 10b?

Line 419: It isn't clear to me how this suggests the presence of an ice layer in the same place following drainage in the previous season? Is there evidence of this in satellite imagery?

Line 425: I suggest slightly rewording the first part for clarity: '(a) Sentinel-2 image acquired on May 14th 2019, with ICESat-2 RGT 727 (occurring on May 15th 425, 2019 and August 14th, 2019) overlapping in green (…). (b) Lake depth derived from Watta (Panel c) and Sentinel imagery (Panel a).' Please also add full stops to make it easier for the reader to separate descriptions of (a), (b), (c), (d) and (e).

Line 433: Refer to Fig. 11d here, and Fig. 11e-i in the next sentence.

Line 436: I agree that the stream is clearly incised on September 24$^{th}$ but think it might be difficult to conclude the same from the April 20$^{th}$ Skysat image.

Line 440: I think it would be helpful here to cite some work showing how surface relief preconditions the spatial distributions of lakes and surface drainage, e.g. Ignéczi et al. (2018).

Line 447: Specify again that the intense melt season was in 2019.

Line 450: Specify Landsat geolocational accuracy (5 m).

Line 460: I suggest citing some other recent studies that have quantified the seasonal evolution of surface meltwater in Antarctica: Dell et al. (2020), Moussavi et al. (2020).

Line 469: See general comment above about data availability.

**Technical/minor corrections:**

Line 12: 'bathymmetric' spelling error (same on Line 57 and 60).

Line 16: Italicise '*in situ*' (and please check throughout).

Line 23: Add comma after '(both publicly-available and commercial)'.

Line 30: This should be Slater *et al.* (2018) (please also check similar instances throughout – especially in places where the reference is unclear e.g. Pope, 2016 or Pope et al., 2016). Some references are also missing from the reference list (e.g. Fair et al., 2020) – please check.

Line 35: Consider rewording to 'led to unprecedented summer mass loss'.

Line 51: Change 'LandSat' to 'Landsat 7 and 8'.

Line 62: I suggest moving '(supraglacial lake depth)' to Line 57 i.e. 'empirical (supraglacial lake depth) bathymetric methods'.

Line 65: Typo, 'wen' should be 'when'.

Line 112: No need to hyphenate 'high spatial'.

Line 119: Replace 'is' with 'are' ('frequently captured multiple times').

Line 120: Specify 'spectral' response curves and write near infrared (NIR) in full here.

Line 135: Comma should be full-stop.

Line 137: Change 'is' to 'are'.

Line 176: Remove duplicate word 'outliers'.

Line 205: New sentence after 'lake edges'.

Line 215: 'in order to exclude regions with moving surface water, which evolves rapidly and can be mistaken for fixed topography'.

Line 218: Remove double comma.

Line 242: Keep to past tense for consistency.

Line 292: Typo ('there were').

Line 435: Missing word ('a' substantial quantity of liquid water).

Line 438: 'connects' to an efficient drainage system.

**References**

Banwell, A.F, Caballero, M, et al. (2014) Supraglacial lakes on the Larsen B ice shelf, Antarctica, and at Paakitsoq, West Greenland: a comparative study. Annals of Glaciology 55(66), doi: 10.3189/2014AoG66A049.

Ignéczi, A, Sole, A, Livingstone, S.J., Ng, F.S, Yang, K (2018) Greenland Ice Sheet Surface Topography and Drainage Structure Controlled by the Transfer of Basal Variability. Frontiers in Earth Science, https://doi.org/10.3389/feart.2018.00101.

Legleiter, C. J., Tedesco, M., Smith, L. C., Behar, A. E., and Overstreet, B. T.: Mapping the bathymetry of supraglacial lakes and streams on the Greenland ice sheet using field measurements and high-resolution satellite images, The Cryosphere, 8, 215–228, https://doi.org/10.5194/tc-8-215-2014, 2014).

Pope, A (2016) Reproducibly estimating and evaluating supraglacial lake depth with Landsat 8 and other multispectral sensors. Earth and Space Science, 3, 176–188, doi:10.1002/2015EA000125.

Pope, A, Scambos T.A et al. (2016) Estimating supraglacial lake depth in West Greenland using Landsat 8 and comparison with other multispectral methods. The Cryosphere, 10, 15-27, doi:10.5194/tc-10-15-2016.

Tedesco, M and Fettweis, X (2020): Unprecedented atmospheric conditions (1948–2019) drive the 2019 exceptional melting season over the Greenland ice sheet. The Cryosphere, 14, 1209-1223, https://doi.org/10.5194/tc-14-1209-2020.

Saunier, S (2020) TN on Quality Assessment for PlanetScope (DOVE) https://earth.esa.int/eogateway/documents/20142/1305226/EDAP-REP-007-TN-on-Quality-Assessment-for-PlanetScope-DOVE-v1.2.pdf

Sneed, W.A and Hamilton, G.S (2007) Evolution of melt pond volume on the surface of the Greenland Ice Sheet. Geophysical Research Letters 34(3), L03501 (doi: 10.1029/2006GL028697).

---

## Author Comment (AC1)

Original Comments (Specific to the Critique)
*In general, this is a well-written manuscript and most of my comments are relatively minor. I would like to see in places some additional detail around the discussion of lake depth retrieval methods (see specific comments).*

*Lastly, The Cryosphere's data policy states that "Authors are required to provide a statement on how their underlying research data can be accessed. This must be placed as the section "Data availability" at the end of the manuscript." Although the authors state at the end of the manuscript that the Matlab code will be converted and shared publicly, I would like to see this section added including statements of how Landsat, Sentinel and Planet imagery can be accessed.*

In addition to the minor comments, we will be releasing the matlab code for Watta within github. In response to both this critique and the other reviewer, we have expanded upon the section relating to lake/ice layers detected by Watta and imagery over both Lake Julian and Lake Ayse. As requested, we have also expanded the discussion about the assumptions made about the surface winds and lake bottom structure.

Partially in response to this critique, we have detailed specific imagery IDs and ICESat-2 track boundaries attached to each scene/depth estimate. This should allow others to easily replicate the dataset in the future, which is unique because of its coincidence with the Operation IceBridge mission. In particular, this dataset will now be used within a classroom by another colleague who works with surface melt estimates. Additionally, all references have been added/checked/removed according to the reviewer's suggestions. Data Availability now includes links for Watta code, Planet imagery, Sentinel-2, Landsat as well as ICESat-2 ATL03.

**Specific comments:**
*Line 17: Specify Landsat 8 OLI, as you are specifying Sentinel-2.*
Altered

*Line 38: 'a common feature on large parts of the ice sheets' – this should be ice sheet (singular) as the paragraph has only discussed Greenland so far.*
Altered

*Line 40: ' (...) of both ice sheets'. So far, Antarctica has not been mentioned in the text. Perhaps add an additional 1-2 sentences in the previous paragraph introducing meltwater production and Antarctic-wide surface meltwater and supraglacial lakes.*
Added sentence about Antarctic ice shelf hydrology

Line 43: Suggest rewording to: 'The complex links between (…) can potentially be deduced (…)'.
Done

*Line 45: Specify here that feature types are 'supraglacial'.*
Done

*Line 46-7: I think detail could be added here outlining both the physically-based and empirically-based methods, together with a slight clarification of the wording, as the physically-based approach has been applied to other optical imagery such as Aster (Sneed and Hamilton, 2007), Landsat (Banwell et al., 2014) and Sentinel. It should also be added to the text that the physically-based method assumes a minimal impact of wind-driven surface waves, low slopes of lake bottoms and a homogeneous lake-bottom albedo on lake depths, which may be particularly important in Greenland (Sneed and Hamilton, 2011). I believe the empirically-based method was first applied to WorldView2 by Legleigter et al. (2014), so I suggest citing this work too (see below for full reference).*
We have added the details requested and the associated reference, although we note that the earliest reference to the empirical method is Box and Ski (2007)

*Line 49: Can you include a specific lake depth limit here? I believe Pope et al. (2016) found that the maximum lake depth that could be derived from the empirical depth retrieval method using in-situ estimates was 5 m.*
Added.

*Line 55: Specify resolution here (e.g. ≤ 3 m, daily) to demonstrate improvements over Sentinel-2.*
Added

*Line 61: Could you specify by how much the physically-based estimates tended to underestimate lake depth?*
Altered "tended to underestimate lake depth" to "tended to underestimate lake depth by over 2m"

*Line 65: Specify native resolution (0.7 m).*
Done

*Line 68-70: This sentence is slightly hard to follow – adding 'from' before 'multiple imagery sources' may improve the clarity?*
Done

*Line 72: It is not clear to me what the representative sections are that are referred to in this sentence – is this part needed?*

Took out the word "representative"

*Line 75: Can you cite any work here to support that this was an unusually intense melt season, e.g. Tedesco and Fettweis (2020)? I would also specify 2019 here too, just to clarify to the reader.*
Added reference

*Line 78: I think this is the first place CAMBOT is used as an acronym, so include the full name (Continuous Airborne Mapping By Optical Translator).*
Added

*Line 85 (Figure 1): Please make the scale bar in the top left panel bigger, and add lat-lon labels to the main panel. Perhaps also add '(A)' to the Lake Ayse label to make clearer how the Skysat image relates to the main panel. In the Figure caption, specify that RGT = repeat ground track. Finally, a small comment but maybe call SkySat boxes 'grey' rather than white in the caption so that it is clear when looking at the main panel*
Altered accordingly, although graticule added in bottom left panel as either text or a graticule in the main panel made the panel difficult to understand. Additionally, latitude/longitude of all lakes has been added to Supplemental Table 1

*Line 100: Specify ICESat-2 confidence levels (i.e. low, medium, high) in brackets.*
Done

*Line 116: Are there any studies you can cite here to demonstrate the PlanetScope radiometry issues? Possibly Saunier (2020)?*
Added.
The improvements to radiometry were being discussed and then implemented at Planet when the work was being conducted, thus I have added the reference but avoided adding more details.

*Line 121: What is the vertical accuracy of the GIMP-2 DEM? Also, include the GIMP-2 dataset citation here (Howat et al.).*
Have added citation to Howat et al., 2014 and a phrase describing the vertical accuracy (as caompared to ICESat) as between ±1m over most ice surfaces and ±30m in regions with high relief.

*Line 143: Specify how lake boundaries are delineated? Discussion of NDWI does not come until Section 3.2, consider moving some of this into the Methods section.*
We elected to reference section 3.2, where the image processing is discussed in greater detail (including how lake boundaries are delineated).

*Line 195: I wonder if you have considered what the effect is of wind-driven scattering on lake surface roughness and on the surface photon return?*
This is absolutely an issue that can drive surface roughness and, I think, an interesting direction for research, but this is beyond the scope of this study.

*Line 218: Specify that NDWIice uses the blue and red bands.*
Added

*Line 220: Is 'limitations' a better word here, rather than 'limits'?*
Agreed.

*Line 246: I think Banwell et al. (2019) calculate a lake-bottom ablation rate of 20.3 mm day-1 on McMurdo Ice Shelf. Perhaps worth adding in?*

We think this may be unnecessary as this is on an ice shelf with rather different dynamics than the lakes studied here. A thorough study of lake ablation would, I think be outside the scope here (and the authors' expertise).

*Line 315: Be explicit in the Figure 5 caption that G = green and R = red in panels c-j.*
Added text accordingly.

*Line 330: In Fig. 6 caption, write abbreviations in full again for clarity (*
Agreed. Text added accordingly.

*Line 345: 'Liquid water (…) remains constant at around 3%' – over what period? Looking at Figure S4, the surface water extent appears to increase?*
Actually, interestingly, that's not the case (which was surprising, hence the inclusion in the text). It is simply distributed differently, as detailed in the next sentence. We have, however, specified the time period as requested for completeness.

*Line 346: Can you refer to a figure/results here to support this statement?*
This now directly reffers to Supplemental Fig. S4. The support to this statement is exactly what the reviewer has observed contrasted with the actual calculation of NDWI values over the ice sheet, namely that there is more surface water over recognizable (larger) features later in the season.

*Line 355: Could you add lake outlines to Panel b of Figure 7?*
Unfortunately, doing so obfuscated some of the contrast in the elevation detail.

*Lines 360-62: Refer to Fig. 8(j-l) end of first sentence, and (m-o) in the next sentence.*
Added

*Line 383: Refer to Fig. 8m-n at the end of this sentence.*
Added

*Line 390: Looking at the SkySat image in Figure 9b, incised streams are also visible to the right of the image; could Lake Julian not also have drained through these?*
It's a possibility (and will include language accordingly), but the timing of the new incised stream and the sudden rapid drainage suggests that this was the most likely cause.

*Line 401: Should this refer to Fig. 10c here, not 10b?*
That's correct. Altered accordingly.

*Line 419: It isn't clear to me how this suggests the presence of an ice layer in the same place following drainage in the previous season? Is there evidence of this in satellite imagery?*
In this case, the floating ice layer shown in the May 14th image is, itself, the evidence that this is the point of drainage (i.e. that it formed and then settled in this location following drainage). This interpretation was corroborated as plausible by several experts on Greenland surface hydrology (including Lauren Andrews, acknowledged here). This is obviously not clear to the reader though so we have altered text to add "that an ice layer had formed at the same place **and settled in this location** following drainage" to make this explicit.

*Line 425: I suggest slightly rewording the first part for clarity: '(a) Sentinel-2 image acquired on May 14th 2019, with ICESat-2 RGT 727 (occurring on May 15th 425, 2019 and August 14th, 2019) overlapping in green (…). (b) Lake depth derived from Watta (Panel c) and Sentinel imagery (Panel a).' Please also add full stops to make it easier for the reader to separate descriptions of (a), (b), (c), (d) and (e).*
Added. This figure has also been altered slightly to more clearly depict the points on imagery.

*Line 433: Refer to Fig. 11d here, and Fig. 11e-i in the next sentence.*
This doesn't match with the figure, but have added specific references.

*Line 436: I agree that the stream is clearly incised on September 24th but think it might be difficult to conclude the same from the April 20th Skysat image.*
This is due to the resolution of the image. We have altered the figure to include an inset panel to focus on the incision point on April 20th.

*Line 440: I think it would be helpful here to cite some work showing how surface relief preconditions the spatial distributions of lakes and surface dradinage, e.g. Ignéczi et al. (2018).*
Have added a sentence with the suggested reference

*Line 447: Specify again that the intense melt season was in 2019.*
Added.

*Line 450: Specify Landsat geolocational accuracy (5 m).*
Added

*Line 460: I suggest citing some other recent studies that have quantified the seasonal evolution of surface meltwater in Antarctica: Dell et al. (2020), Moussavi et al. (2020).*
Moussavi et al., 2020 is mentioned several times earlier in the paper, but we have mentioned Dell et al., 2020 specifically pointing out the feature-tracking capabilities

*Line 469: See general comment above about data availability.*
Addressed

**Technical/minor corrections:  ALL ADDRESSED except as noted**
*Line 12: 'bathymmetric' spelling error (same on Line 57 and 60).*
*Line 16: Italicise 'in situ' (and please check throughout).*
*Line 23: Add comma after '(both publicly-available and commercial)'.*
*Line 30: This should be Slater et al. (2018) (please also check similar instances throughout – especially in places where the reference is unclear e.g. Pope, 2016 or Pope et al., 2016). Some references are also missing from the reference list (e.g. Fair et al., 2020) – please check.*
*Line 35: Consider rewording to 'led to unprecedented summer mass loss'.*
**This does not flow with the clause "in the past 50 years" unfortunately**

*Line 51: Change 'LandSat' to 'Landsat 7 and 8'.*
*Line 62: I suggest moving '(supraglacial lake depth)' to Line 57 i.e. 'empirical (supraglacial lake depth) bathymetric methods'.*
*Line 65: Typo, 'wen' should be 'when'*
*Line 112: No need to hyphenate 'high spatial'*
*Line 119: Replace 'is' with 'are' ('frequently captured multiple times').*
*Line 120: Specify 'spectral' response curves and write near infrared (NIR) in full here.*
*Line 135: Comma should be full-stop.*
*Line 137: Change 'is' to 'are'.*
*Line 176: Remove duplicate word 'outliers'.*
*Line 205: New sentence after 'lake edges'.*
*This is a new sentence*
*Line 215: 'in order to exclude regions with moving surface water, which evolves rapidly and can be mistaken for fixed topography'.*
*Line 218: Remove double comma.*
*Line 242: Keep to past tense for consistency.*
*Line 292: Typo ('there were').*
*Line 435: Missing word ('a' substantial quantity of liquid water).*
*Line 438: 'connects' to an efficient drainage system.*

---

## Author Comment (AC2)

**Major concerns:**

1. *Lack of evidence for sub-surface ice layers: The manuscript often refers to ice cover on melt lakes as well as sub-surface ice layers without showing evidence that such features are indeed present. In fact, most of the features that are described as "sub-surface ice layer" very well match the appearance of "second return" artefacts that result from the ATLAS sensor being saturated due to specular returns from flat surfaces, and many photons returning to the instrument during its dead-time without being detected. This issue is also briefly described in Martino et al. (2020), and further detailed in the "Specular Returns" section of the known issues document for the ATL03 product. I would highly recommend that the authors consult these two sources and decide whether they are still convinced that these second returns are signals from sub-surface ice layers rather than just artefacts in the data. If so, I would like to see convincing evidence for the claimed widespread existence of such sub-surface ice layers in a revised manuscript.*

We thank the reviewer for this insight and have examined both texts suggested. We had, earlier, discussed the specific case of Lake Ayse with Dr. Martino when this work was being developed and proceeded with the code because this did \*not\* seem to be a case of specular returns or instrument echo. However, the case for Lake Julian is a bit more ambiguous. The subsurface ice layer detection is intended to locate ice layers, but will definitely also capture both of phenomena described. We have added text (as follows) to make a less bold claim than in the original and also underline that the detection of ice layers is inclusive of both cases we deem to be real as well as specular returns/instrument echo. Additionally, we have included imagery (which is not covered by labels) in Fig. 10 which will hopefully make apparent why we are interpreting these physical structures the way that we are.

In Section 3.1, introducing the issue and relating it to Lake Ayse:
Two potential sources of ambiguity with the subsurface ice classification are: (a) the possibility for specular returns and (g) apparent multiple surface returns which resulting from instrument echo. Specular returns over flat water (implying high energy return), return a strong surface as well as multiple layers below the surface spaced according to the ATLAS deadtime (1m below the surface and a potential tertiary return below that). Echoes produced by electronic noise in the instrument, which also frequently occur very smooth water surfaces, can similarly produce a strong return at the surface with double echoes at ~2.3m and ~4.2m below the surface. (Martino et al., 2020). The categorization of subsurface ice (as in with Lake Ayşe in Fig. 3) are reliant on visual inspection. In this case, we assume subsurface ice because the layer is less than 1m from the surface and shows trailing photons towards a weakly-resolved lake bottom rather than a distinctive sharp horizontal layer with no curved bottom return. If this were a specular return, we would expect a high energy surface return to obfuscate the lake bottom entirely.

In Section 5.2, discussing drainage points of Lake Julian:
The *Watta* designation of "ice surface" here is likely, but not unambiguous, as this method will capture both real ice surface and false dual returns as detailed in Section 3.1. The main attributes typical of the false dual returns are a strong top surface over surface water in a flat region, followed by weaker returns at predictable intervals (~1m and ~1m for the specular return, ~2.3 and ~4.2m for the instrument echo). In this case, a specular return would be the most likely cause for a false dual-return due the spacing. However, we first note that the surface in this region is not flat and we do not see the predicted strong surface return followed by a weaker echo ("surface" and "ice" layers are of equal thickness). Additionally (a) in the case of Pt. B, the top layer contains no dual return (b) in the case of C, a distinct gap occurs in the surface. Both of these correspond to ice/water in imagery. For Pt A and Pt D, imagery suggests that these points occur at a convergence of streams. These could, however be either water or ice as no distinctive bottom return is detected. With the current available information, the "ice surface" detection

will still require manual inspection; future improvements to the code may account for the known issue with false dual returns as knowledge in this area develops.

2. *Insufficient information to make methods repeatable: One of the main objectives of this manuscript is to present the new Watta algorithm. Yet, the algorithm is not described in much detail, and the information provided is certainly not enough to replicate the results in this study. I think the easiest way to fix this would be to archive the already existing matlab code on a platform such as Zenodo (which was used by the authors to archive the lake depth data set). If the authors do not want to share their code, I think that they should provide some more detailed pseudo code (including a list of parameters and chosen values), or a more detailed text description at the very least. Furthermore, from the information provided in the manuscript, it does not seem possible for readers to check the underlying data themselves. This means that it is very hard to verify any claims made based on "visual inspection". No locations are given for the lakes under consideration, and ICESat-2 data is only referred to by its track number (not the spot, so this could refer to any of the six beams) and plotted against along-track distance with the zero point seemingly arbitrarily chosen. At the very least, I think that the authors need to provide latitude and longitude coordinates for each of the lakes considered in this study and to specify the ICESat-2 spot (GT1L, GT1R, GT2L, GT2R, GT3L, GT3R) for each lake section used with Watta. This information could be included in table S1 in the supplement. Since each of of the lakes has an associated ICESat-2 overpass, this information could be automatically extracted directly from the corresponding ATL03 or ATL06 data by, for example, using the median latitude for each of the segments shown and then querying for the corresponding longitude along the ground track. (Other information useful to readers (spot, beam type, acquisition time, etc.) could easily be printed out at the same time.)*

Accordingly, we will be releasing the current version of *Watta* matlab code to the public. Additionally, we will be adding details about (a) ICESat-2 data, specifically the spot as well as the maximum latitude and longitude of the segment within Supplemental information (b) concurrent imagery IDs from all sources. Together, these should allow users to reproduce these segments with minimal effort.

**Minor concerns / suggestions:**

1. *I think some similar ICESat-2 shallow bathymetry literature should be cited. There are a few articles out there with similar methodology, just applied to satellite- derived bathymetry outside the polar regions. Examples would be Albright and Craig (2020) or Thomas et al. (2020).*

These have been included in the text.

2. *It is unclear to me how the numbered lakes relate to the named lakes. Some sort of explanation for choosing to refer to some lakes by numbers and some by names should be included. A map with the locations of all lakes would be great.*

To clarify, we have added lake names and numbers in Supplementary material. A map with all locations is difficult to make meaningful as the lakes are too small for the total area of each pass. However, we are including specific latitude/longitude boundaries for the lake segments from ICESat-2 (organized by RGT, and therefore able to be mapped to the AOIs present in Figure 1) in order to enable readers to locate and reproduce lakes more easily.

3. *It is not clear to me from the text how geolocation and co-registration relate to each other here. Can you explain how matching the ATL03 point cloud with the GIMP-2 DEM to reduce square error will improve the co-registration between ICESat-2 and Landsat 8 data? To my understanding, GIMP-2 DEM elevations are mostly derived from WorldView stereo imagery, and if there is a significant difference in acquisition time between the image underlying the DEM and the Landsat 8 / ICESat-2 lake observations then surface topography could have changed significantly in the meantime due to ice flow or surface processes. With a geolocation accuracy of roughly 5 meters for both ICESat-2 and Landsat 8, I would assume that simply mapping both datasets to the same CRS would give better results than the intermediate use of DEM elevations. Admittedly, this might be me not fully understanding co-registration, yet it would be nice to be provided with some more detail/explanation, or to see evidence that this intermediate step using DEM elevations actually improves the method in a meaningful way.*

In fact, there was not really any improvement in coregistration using this process (and none that impacted the segment used for the empirical depth estimate). This step was performed nevertheless as a check as Landsat 8 and ICESat-2 geolocation was based on the GIMP-2 DEM (but this process was not transparent to us), and to monitor for anything that looked like large-scale deviation from the DEM due to changes in ice flow (although this was minimal as lakes largely conformed to bed topography. We decided to keep this step in the workflow to potentially use imagery sources in the future which were geolocated to another DEM.

Regarding temporal changes in surface topography, we note that the portion of the ICESat-2 beams that were used for coregistration were long enough to incorporate large-scale relief (which would not be affected by ice flow).

Admittedly, perfect geolocation of high-resolution imagery given ice flow is a challenge with this work, and to our knowledge, this is ongoing research for other groups.

4. *The results shown on the figures could be made somewhat more accessible to readers: I would suggest to plot any ATL03 data with latitude on the horizontal axis while also including a scale bar for along-track distance. This, along with the information about which ICESat-2 track and spot is shown on which date, would already be enough for readers to figure out where to find all the underlying data. Plotting ICESat-2 ground tracks on top of imagery or image-based depth estimated wherever applicable would help readers with visual verification of some of the claims made in the text. A graticule on some maps/imagery would help as well.*

To address the issue of repeatability, we are detailing the specific location of lakes (latitude/longitude boundaries of ICESat-2 segments used) in Supplemental Table 1 in addition specific concurrent imagery identifiers used for each calculation. We are avoiding including a full catalogue of imagery and overlying ICESat-2 tracks for two reasons (1) the overlying track tends to obfuscate the image substantially (2) the resulting file size (for sufficient resolution of imagery) is very large given all of the lakes in question.

The choice of axes was actually a specific request made by audience members within a previous presentation of this material. As much of the focus of this paper is related to small-scale features, we chose this axis to allow the reader to easily understand the length of lakes or ice cover.

**Line-by-line comments:**

*Line 14: From just reading the abstract it is unclear what is meant by "corrected" depth. Line 17: Landsat 8*
Altered to depth corrected for refraction

*Line 18: You are stating 46 lakes, but I am counting 45 lakes in table S1, plus the five named lakes in Figure 1 for a total of 50 lakes? Line 22: please spell out CAMBOT the first time you use it (Continuous Airborne Mapping By Optical Translator)*
The additional lake did not have coinciding imagery. In short: the 46[th] lake was used in Watta development, but only 45 had overlaying imagery with which to extract lake volumes.
We spelled out CAMBOT on its first occurance.

*Line 40: You mention "both ice sheets" here, yet surface melt in Antarctica has not been discussed. While this paper only uses data over Greenland, I think it would be beneficial to briefly mention surface melt in Antarctica and how it is believed to be connected to ice shelf disintegration via hydrofracture.*
A sentence in the following paragraph now discusses Antarctic ice sheets

*Line 54: Since you specify for Sentinel-2, can you specify what the "higher spatial and temporal resolution" is?*

Sentence has been altered from:
Commercial satellite imagery, which is poised to expand
substantially in the future, can help fill the gap in coverage of small-scale melt and melt-induced features at a higher spatial and temporal resolution, complementing estimates resolved from Sentinel-2.
To:
Commercial satellite imagery, which is poised to expand substantially in the future, can help fill the gap in coverage of small-scale melt and melt-induced features at a higher spatial (<3m) and temporal (multiple daily passes) resolution, complementing estimates resolved from Sentinel-2.

*Line 56: This sounds very wordy. Could simply say "ICEsat-2 now makes it possible to replace…"*
Altered accordingly

*Line 60: typo (bathymmetry → bathymetry)*
Corrected (here and in other locations)

*Line 64: I am sceptical about the presence of ice layers under the water surface. See above in the major concerns section.*
Addressed separately (see above)

*Line 64: Maybe here I would specify that by "the native resolution of the ATL03 photon cloud" you mean the 0.7m laser pulse frequency in along-track distance*
added

*Line 65: typo (wen → when)*
fixed

*Line 85 / Figure 1: Can you add latitude and longitude labels, or preferably a graticule in the right panel? Please also specify in the caption that RGT = ICESat-2 "Reference Ground Track", not "repeat ground track". The ground tracks that should be repeated (in the polar regions) are the six spots GT1L,*

*GT1R, GT2L, GT2R, GT3L and GT3R for each numbered track. The RGT should be the point directly at the nadir, so unless ATLAS is pointing off-nadir it should fall right between GT2L and GT2R.*
Corrected as requested, but with graticule added to the bottom left panel (as the addition in the main panel created too much visual noise). However, in order to better identify the locations of specific lakes, we will be adding max/min lat/lon values to Supplemental Table 1 (in addition to imagery identifiers associated with each lake). This should also address concerns about repeatability.

*Line 92: This paper is largely about ICESat-2 so you might want to spell it out: "Ice, Cloud, and Land Elevation Satellite" and possibly ATLAS = Advanced Topographic Laser Altimeter System*
Altered

*Line 98: It is unclear to me what you mean by "using x signal photons per shot". Are you referring to the expected number of signal photons that ATLAS will detect per pulse? Are these values over land ice? Is there a citation for these values?*
These are from the ICESat-2 science specs https://icesat-2.gsfc.nasa.gov/science/specs, also in Neumann et al., 2019 (cited)

*Line 101: typo (MacGruder → Magruder)*
Altered

*Line 114: TOA has not been defined before → top of atmosphere*
Altered

*Line 121: It is unclear to me here what the role of the DEM is in geolocation. Line 124: "each area was approximately on average" makes no sense?*
Altered (deleted "on average") and a direct reference to the Imagery Processing section is included (as this is a bit difficult to summarize)

*Line 137: There are only 45 lakes in the supplemental table?*
This is because only 45 lakes had both coincident imagery and ICESat-2. One lake was used for the development of Watta, but did not have coincident imagery that was usable (the lake drained too quickly afterwards). We have corrected the abstract accordingly.

*Line 150: I think it might be good to point out somewhere that if the empirical estimates are "time, location and sensor specific", then your method is currently limited to producing valid depth estimates for imagery scenes that overlap with an ICESat-2 overpass over a melt lake within that scene and a three-day window. This is a limitation that the physical models don't have.*
This is a fair point and has been explicitly noted around Line 77

*Line 166: How are "outliers" detected?*
We have added the following text to clarify: "whereby the number of standard deviations used to detect an outlier and the number of photons used to calculate a mean (window) increase with over several steps"

*Line 191: What you describe here sounds exactly like artefacts in the data that come from ATLAS's dead-time when the sensor is oversaturated by a specular return. If you really believe that this is sub-surface ice in some cases, then I would need to see evidence for that to be convinced. (see major concerns section)*
Addressed in the main response.

*Line 201: missing full stop after "lake edges"*
Fixed

*Lines 205-208: It is not clear from the text how matching the ATL03 point cloud with the GIMP-2 DEM to reduce square error will improve the co-registration between ICESat-2 and Landsat 8 data. GIMP-2 DEM elevations are mostly derived from WorldView stereo imagery, and if there is a significant difference in acquisition time between the image underlying the DEM and the Landsat 8 / ICESat-2 lake observations then surface topography could have changed significantly in the meantime due to ice flow or surface processes. With a geolocation accuracy of roughly 5 meters for both ICESat-2 and Landsat 8, I would assume that simply mapping both datasets to the same CRS would give a better co-registration than the intermediate use of DEM elevations. If this is not the case, it would be nice to see some sort of proof that this intermediate step using DEM elevations actually improves coregistration.*

In fact, there was not really any improvement in coregistration using this process (and none that impacted the segment used for the empirical depth estimate). This step was performed nevertheless as a check as Landsat 8 and ICESat-2 geolocation was based on the GIMP-2 DEM (but this process was not transparent to us), and to monitor for anything that looked like large-scale deviation from the DEM due to changes in ice flow (although this was minimal as lakes largely conformed to bed topography. We decided to keep this step in the workflow to potentially use imagery sources in the future which were geolocated to another DEM.

Regarding temporal changes in surface topography, we note that the portion of the ICESat-2 beams that were used for coregistration were long enough to incorporate large-scale relief (which would not be affected by ice flow).

Admittedly, perfect geolocation of high-resolution imagery given ice flow is a challenge with this work, and to our knowledge, this is ongoing research for other groups.

*Line 208: Do you mean a margin of 0.2 degrees in latitude and/or longitude?*
The reference was to latitude, and text has been added accordingly

*Line 217: two commas after NDWI_ice, missing full stop before "Boundaries".*
Altered

*Line 224: typo (MacGruder → Magruder)*
Altered

*LIne 225-226: "a line 6m in each direction perpendicular to the ICESat-2 beam" seems like a rather confusing way to describe a circle of 6 m radius around the location of the photon.*
Indeed! Altered accordingly.

*Line 230 / Figure 3: Please plot the ground track of the ATL03 data shown in the top left panel on top of the depth estimates shown in the bottom left panel. Please spell out "Elevation" and "Along-track distance" in the top left panel. Also, why is along-track distance going from roughly -50 m to 800 m? I think the ICESat-2 convention is that along-track distance is measured from the last equator crossing? It would probably be more helpful for the reader if elevation was plotted against latitude, with a scale bar indicating along-track distance.*
The choice of axes was actually a specific request made by audience members within a previous presentation of this material. As much of the focus of this paper is related to small-scale features, we chose this axis to allow the reader to easily understand the length of lakes or ice cover. However, with regard to the location of the lakes, we will be including the latitude/longitude extents of each lake within supplementary material, which should allow for the specific identification of features if desired.

*Line 245: spell out "2" → two*
Altered

*Line 259-260: If performance evaluation is done by "visual inspection", it would be nice if the reader could also get to see a few examples of imagery with precise ICESat-2 ground tracks plotted on top, for their own visual inspection.*
Actually, "visual inspection" here refers to Watta-calculated depths from ATL03 alone, which is largely the objective of Supplemental Figure S3. We have made this more explicit by addition additional depths.

*Line 260: correlation coefficient between what? NDWI and Watta-derived depth?*
Text added to make this explicit

*Line 264-265: "reference ground track (RGT) 1222, Lake 3 in Fig. S4": Should be referring to Fig. S3.*
Altered

*Line 265-266: "the presence of subsurface ice did not always preclude the presence of a strong bottom return" → This suggests to me that it's even more likely that this "subsurface" ice layer might not exist, and that it's actually the sensor saturation and dead-time effect. (see major concerns section)*
Addressed in the main response, although we note that a bottom return was present, it was just somewhat weaker.

*Line 266: "(e.g. Lake 7, RGT 1169, Fig. S4)": Should be referring to Fig. S3. Also, I don't really see anything indicative of subsurface ice in Lake 7, RGT 1169, Fig. S3.*
Addressed in the main response

*Line 279: It sounds like you are using the $R^2$ for performance evaluation of the empirical model, but this would mean to evaluate the model on the data that was used to generate the model in the first place. So it should be made clear that the $R^2$ cannot be considered a performance metric for a model across an entire lake basin, and rather that it merely indicates how well you were able to fit the empirical model to the data along the given ICESat-2 ground track. However, the underlying model is rather simple and based on physics, so overfitting is probably not much of an issue here.*
Additional text has been added to make this more explicit

*Line 292: typo (there're were → there were)*
Altered

*Line 293-294: "future users would be able to select bands or combinations [...] that provide the greatest fidelity to ICESat-2 based observations": I know what you mean by that, but the way it is phrased it sounds like a bulletproof recipe for overfitting the data...*
We have dropped the clause "that provide the greatest fidelity to ICESat-2 observations" to avoid this

*Line 304: Technically GT3L describes a "spot", not a beam. Two beams (one strong, one weak) will alternate in pointing at that particular spot, switching off whenever ICESat-2 performs a yaw flip.*
We have made this clearer by rephrasing as "Over this spot, covered by the 3l beam"

*Line 309: typo (lake → lakes)*
Fixed

*Line 313 / Figure 5: "Sentinel-2 (l,m) and Planet SkySat (n,o)": should be "Sentinel-2 (k,l) and Planet SkySat (m,n)". Also, what does the red box in panel c indicate?*
Altered. Added "Red box in (c)highlights region where underlying crevassing is captured

*Line 328 / Figure 6: Can you show the ICESat-2 ground track on the right panels? It is very hard to see what is going on without that information. Also, it is pretty clear from context what the abbreviations Sent/LSat/SSat/PS/R/G mean here, but at least somewhere you should specify this for clarity. Also please try to stay consistent across all figures. I have noticed images with labels "Sentinel-2", "Sentinel", "Sent" and "S" across the figures in the paper, and they all refer to the same thing.*
We have added an explanation of the abbreviations in the figure caption and altered the designation for "Sent" in other figures to make this a bit more consistent when possible. The ground track over Lake Ayse is shown in Figure 1, which we have made explicit in the figure caption for Figure 6.

*Line 344: You want to refer to Supplemental Fig. S4 here, not S5*
Altered accordingly (here and elsewhere)

*Line 356 / Figure 7: Can you label lake Niels and lake Julian on the left panel?*
Altered accordingly

*Line 407: Ice motion should not be adjusted for in geolocation?*
Because geolocation will fit to Landsat imagery (itself geolocated using the GIMP-2 DEM), we remain reliant on how well Landsat captures ice motion following from the GIMP-2 DEM upon which it's based. It remains possible that minor ice motion will not be perfectly captured (This would require feature-tracking which is outside the scope of this study).

*Lines 409-419: What's shown in cyan in fig 9d does not look like ice cover to me. Also none of the satellite imagery seems to show the presence of ice cover. Can you corroborate your claims about ice layers? From looking at the figure, I would guess those are specular returns from water surfaces. (see also major concerns section)*
Addressed in the Main Concerns above

*Line 425: This is not the reference ground track for track 727. This must be GT1L (based on looking at the data myself), which is roughly 3.3 km offset from the RGT! The big stars used to show the locations A-D very much cover the actual features, which makes it hard to see any of the things discussed in the text. Can you plot the actual precise ground track 727 GT1L for this overpass on panel e as a (very) fine line, and indicate locations A-D with arrows pointing at the features without covering them.*
What is shown is, in fact, the track for gt1l. We have altered the text to clarify this. We are reluctant to add a line to the image in panel (e) as even a fine line obfuscates much of the image. However, we have altered this figure to include two panels (on the two days surrounding the ICESat-2 pass) where the precise location of the apparent dual return is indicated with arrows (thus not obfuscating the imagery at the location).

*Lines 458-463: This paragraph about Antarctica does not fit into the conclusions section. The information about ice shelf stability considerations, etc. would fit nicely into the introduction/background information about surface melt, where "both ice sheets" are already mentioned. Then, in the conclusion section you could just briefly mention that Watta could be used in Antarctica as well.*
As suggested, the text applying to Antarctica has been moved into the Introduction.

*Line 469-470: The goal of implementing Watta in an open-source framework is commendable and would certainly be beneficial to the scientific community. Yet, it would also be helpful to publish the already existing matlab code along with the manuscript. (also likely the easiest way to address my major concern about methods repeatability)*
Addressed in Main Concerns.

---

## Author Response (AR2)

*The revised version of this manuscript resolves all the major and minor concerns I had about the initial version. It is a novel and valuable contribution to the growing scientific literature on remotely sensed depth estimates for supraglacial lakes. I recommend the manuscript for publication subject to the following technical corrections:*

*102: typo - delete "Center track, s"*

Corrected

*107: put a comma or "and" between "instrument" and "distributed"?*

Comma added

*110: you already specified earlier what ATLAS means*

Full name removed

*113: I see now what you mean by signal photos per shot. You might consider something along the lines of replacing "using" with "returning" because the instrument uses many more photos per laser pulse, just that only this few are returned to the sensor as signal photons. You may also replace "shot" with "laser pulse" to conform to the ATL03 ATBD Appendix D (Lexicon), but I'm aware that the ATBD and Technical Specs webpage contradict themselves with that nomenclature so "shot" should be fine as well.*

We adopted both suggestions. The sentence now reads 'with the strong beam returning 0.6-3.9 signal photons per laser pulse vs 0.6-1.0 signal photons per laser pulse for the weak beam'

*150: replace the comma with a full stop before "We discuss"*

Fixed.

*170: "probability of likelihood" is a tautology*

Indeed! We removed 'probability of'.

*179: add spaces between numerical value and unit in "0.1m" and "0.3m" (SI unit style convention; also in other locations later in this manuscript)*

Fixed throughout the manuscript

*221: typo - replace (g) with (b)*

Fixed

*223-224: The ATLAS dead time is ~3 ns, so the second return should be roughly 3 ns * (speed of light) / 2 ≈ 0.45 m below the primary surface. Looking at a few tracks with clear dead time*

*artefacts show that this is indeed the case (e.g. track 848 GT3L on 2019-08-22 between at 79.087<lat<79.127). You can cite Lu et al. (2021, https://agupubs.onlinelibrary.wiley.com/doi/full/10.1029/2021EA001729) on this. However, I am aware that the ATL03 known issues document incorrectly states that the echo would occur around one meter below the surface. So the confusion here is not the fault of the authors, but this number should still be corrected to prevent readers from mistaking sensor artefacts for signals in the data. This will be updated in the ATL03 known issues document for release 005.*

We thank the reviewer for pointing this out. We were confused ourselves about the 1 m, given the 3 ns deadtime, and read the paper by Lu et al. with much interest. A reference to this is now included in our revised manuscript.

*227: "we assume subsurface ice because the layer is less than 1m from the surface": unfortunately, you can't make that argument here, because you would expect the dead-time return to be around half a meter from the surface. Your reasoning for a likely ice layer seems convincing enough even without this remark, so I would just scratch that part.*

Following the reviewer's suggestion, we removed the part about the layer being at less than 1 m below the surface. The last part of the paragraph now read: "In this case, we assume subsurface ice because the layer shows trailing photons towards a weakly-resolved lake bottom rather than a distinctive sharp horizontal layer with no curved bottom return.".

*229: "If this were a specular return, we would expect a high energy surface return to obfuscate the lake bottom entirely." This is not true - as counterexamples see the section of track 848 mentioned above, or the lake over ice example in Lu et al. (2021). Remove this sentence.*

Sentence removed

*252: footprint size seems more relevant here than geolocation accuracy, but the footprint is 13 m in diameter, so a radius of ~6m seems reasonable either way*

OK

*279: supplemental fig S3, not S4*

Corrected

*460: ~0.45 m and possibly ~0.9 m for the specular return*

Corrected

*483: typo: add a space between "lakes" and "are"*

This seems to be Word issue, there actually was a space but barely visible…Fixed!

*512: typo: double comma*

Fixed

In addition to the suggestions of the referee, we also removed a number of typo's, double or missing spaces, and the made the following changes (line numbers refer to the numbering in the document with track changes):

*Line 31*: (IPCC 2019, **Special Report on the Ocean and Cryosphere in a Changing Climate**) -> (IPCC, 2019)

*Line 106*: 'capturing lake depths at various stages of lake development during the summer of 2019, **an unusually intense melt season**.'-> 'capturing lake depths at various stages of lake development during **the unusually intense melt season** of summer of 2019.'

*Lines 135-139*: information added about ICESat-2 beams naming convention: "**Within a single track, the beam pair is designated by a number, i.e. \3" in "gt3r"**. Each beam pair consists of a strong and weak beam, with the strong beam returning 0.6-3.9 signal photons per laser pulse vs 0.6-1.0 signal photons per laser pulse for the weak beam (Neumann et al., 2019). **The beam is designated with "r" or "l", depending on the orientation of the satellite, as in "r" in "gt3r"**."

*Line 159*: **R**eflectance -> **r**eflectance

*Line 189*: Supplemental Table **[1]** -> Supplemental Table **1**

Line 198: *in situ* depth estimates **(D)**

*Line 213-214*: Clarified that 75 photons before and after the individual photons are used. This was ambiguous in the original text: "The Surface Detection module determines, for a collection of photons surrounding any individual photon **(75 collected before and 75 collected after; selected in step a)**."

*Line 224*: 'with' removed: "the number of photons used to calculate a mean (window) increase  over several steps."

*Line 244*: "(e.g. as in Fig. 2 step a)" -> "(e.g. as **for the lake shown** in Fig. 2 step a)"

*Line 340-341*: "are shown in Supplemental Fig. S4, Table S1." -> "are shown in Supplemental Fig. S4 **and with their coordinates and relevant statistics listed in** Table S1."

Line 454: "(Supplemental Fig. S4)" -> "(**see also** Supplemental Fig. S4),"

*Line 478-480*: "with imagery below a 1 meter resolution" -> "with imagery **at a 1 meter resolution or below**"

*Line 523-524*: repetition of "slight" removed: "slight ice motion, e.g. a **slight** southward shift" -> "slight ice motion, e.g. a **small** southward shift"

*Line 535*: "These are shown in cyan in **Fig. 9d**" -> "These are shown in cyan in **Fig. 10d**"

*Lines 559-561*: "Pt." replaced by "point"

*Line 617*: "**matlab**" -> "**MATLAB**"

*Line 620*: author contribution added

---

## Author Response (AR3)

Dear Dr. Sørensen,

Thank you for your reply. We've attached the corrected manuscript with the requested corrections (as below), but regarding L. 240, the correction to the reference to the footprint size required a new reference. Since the original manuscript was written, there has been a new publication regarding footprint size of ICESat-2 and we have accordingly switched out the original reference to Magruder et al., 2020 with Magruder et al., 2021.

L. 34: do you mean 1970s/1980s ?
L. 66: lake- bottom. -> lake-bottom ?
L. 82: The ICESat-2 laser altimeter -> The ICESat-2 laser altimeter data
L. 134: \3" -> "3"
L. 251: return -> returns
L. 240: I believe that the referee pointed out that maybe it is more relevant to state here what the footprint size is, than what the geolocation accuracy is. It doesn't affect your choice of radius, though.

Thank you again for your attention.

Sincerely,

R. Tri Datta
Bert Wouters